# FEDPOB: SAMPLE-EFFICIENT FEDERATED PROMPT OPTIMIZATION VIA BANDITS

## ABSTRACT

The performance of large language models (LLMs) is highly sensitive to the input prompt, making prompt optimization a critical task. However, real-world application is hindered by three major challenges: (1) the black-box nature of powerful proprietary LLMs, (2) the need for high sample efficiency due to query costs, and (3) the desire for privacy-preserving collaboration among multiple users. To address these challenges simultaneously, we introduce a novel framework for sample-efficient federated prompt optimization based on multi-armed bandits (MABs). The MAB framework is uniquely suited for this problem as it is (1) inherently a black-box optimization method, (2) practically sample-efficient, and (3) enables collaborative learning with theoretically guaranteed benefit from more participating agents. We first propose the *Federated Prompt Optimization via Bandits* (FedPOB) algorithm, a federated variant of the Linear UCB algorithm, where agents collaborate by sharing model parameters instead of raw data. We then extend our approach to the practical setting of comparative user feedback by introducing *FedPOB with Preference Feedback* (FedPOB-Pref), an efficient algorithm based on federated dueling bandits. Extensive experiments demonstrate that both FedPOB and FedPOB-Pref significantly outperform existing baselines and that their performance consistently improves as more agents participate in the collaboration, validating the effectiveness of our federated approach.

## 1 INTRODUCTION

Large language models (LLMs) have achieved impressive performance in a variety of real-world applications (Guo et al., 2025). However, the performance of LLMs has been shown to be highly sensitive to the input *prompt* (Zhou et al., 2023; Lin et al., 2024b). Consequently, *prompt optimization*, in which we aim to find the best prompt for a task, has emerged as a critical research area. Despite its growing popularity, the widespread real-world adoption of prompt optimization is still hindered by three important challenges.

The first challenge is **black-box access**. Some of the most powerful LLMs, such as ChatGPT and Gemini (OpenAI, 2023b; Team et al., 2023), are proprietary, black-box models that are only accessible via API queries. This limited access creates an immense challenge to prompt optimization. The second challenge is **sample efficiency**. Since querying powerful LLMs is often costly in both time and financial resources, it is of paramount importance to develop methods that can identify the optimal prompt for a given task using a small number of interactions. The third challenge is enabling **collaboration** among multiple users. As LLMs become more widely adopted, a natural and important question arises: how can multiple users, each with their own prompt optimization tasks, collaborate to accelerate their progress? A key constraint in such a collaborative setting is user privacy, as participants are typically unwilling to share their proprietary data, such as the history of tested prompts and their corresponding performance scores. This scenario naturally aligns with the principles of *federated learning* (FL) (Kairouz et al., 2019; McMahan et al., 2017), where distributed agents collaborate on their machine learning tasks without exposing their raw data. As an example, federated prompt optimization enables multiple hospitals to collaboratively learn improved prompts for LLM-assisted tasks (e.g., generating diagnosis suggestions) without sharing sensitive patient data due to privacy regulations. Similarly, it allows mobile users to collaboratively learn improved prompts for on-device assistant tasks (e.g., drafting personal emails or managing schedules) without exposing their private interaction history.

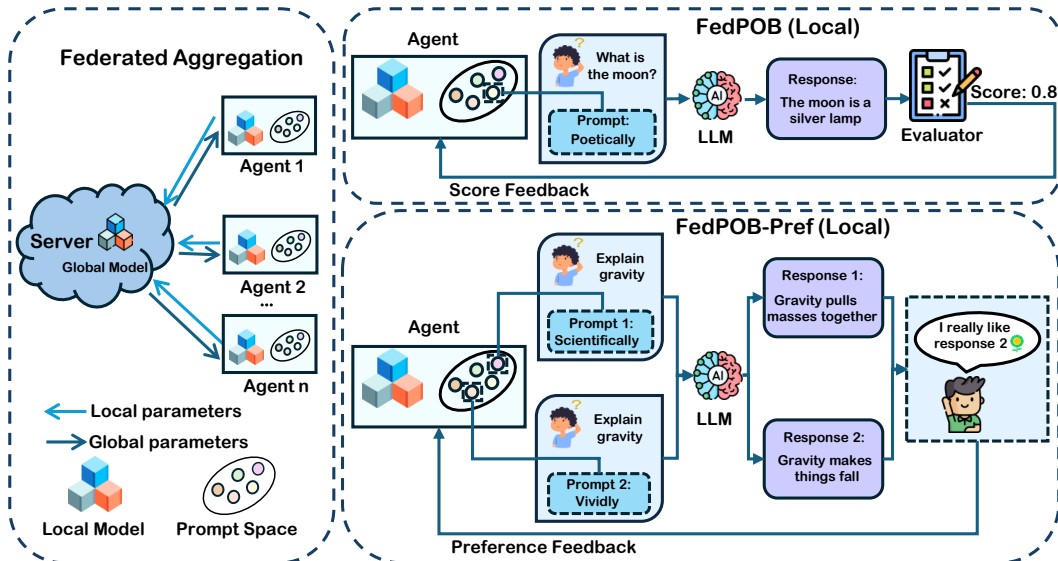

Figure 1: An overview of our proposed federated prompt optimization frameworks. `FedPOB` handles direct score feedback, while `FedPOB-Pref` is designed for pairwise preference feedback.

To tackle the combined challenges of black-box access, sample efficiency and privacy-preserving collaboration, we propose a new class of federated prompt optimization algorithms built upon the *multi-armed bandit* (MAB) framework (Lattimore & Szepesvári, 2020). MABs are exceptionally well-suited for this problem for three main reasons. First, MAB algorithms do not require gradient information and are inherently **black-box optimization methods**. Second, they are designed to efficiently balance the exploration-exploitation trade-off, enabling them to solve complex black-box optimization problems in a **sample-efficient** manner, a property that has been successfully leveraged in recent work on prompt optimization (Lin et al., 2024b; Wu et al., 2024). Thirdly, federated MAB algorithms (Shi & Shen, 2021; Dubey & Pentland, 2020; Dai et al., 2023) provide strong theoretical guarantees, ensuring that **performance improves as more agents participate in the collaboration** (Wang et al., 2020).

Our first contribution is the *Federated Prompt Optimization via Bandits* (`FedPOB`) algorithm. This method is based on a federated variant of the classic Linear Upper Confidence Bound (LinUCB) algorithm (Abbasi-Yadkori et al., 2011; Wang et al., 2020). In our `FedPOB` algorithm, each agent utilizes a pre-trained embedding model to represent the prompts and a linear model to predict their performance. Collaboration is achieved by having agents periodically exchange and aggregate their LinUCB parameters, thereby learning from the collective experience of all agents without requiring them to share any sensitive raw data. Importantly, thanks to the solid theoretical guarantees of the federated LinUCB algorithm (Wang et al., 2020), the performance of our `FedPOB` algorithm is theoretically guaranteed to improve with a larger number of collaborating agents.

In addition, we consider the highly practical setting of prompt optimization with *preference feedback*, where explicit performance scores are unavailable and we are only able to observe relative preference feedback (e.g., the user prefers the response from prompt A than that from prompt B). This problem was introduced by Lin et al. (2024a) to address scenarios where user feedback is inherently comparative. To enable sample-efficient federated prompt optimization in this novel setting, we introduce our second algorithm, *FedPOB with Preference Feedback* (`FedPOB-Pref`). This algorithm is a practical adaptation and modification of the federated linear dueling bandit framework proposed by Huang et al. (2025). Specifically, our `FedPOB-Pref` algorithm significantly reduces the communication complexity of the methods from Huang et al. (2025) while maintaining the strong empirical performance. An overview of both `FedPOB` and `FedPOB-Pref` is illustrated in Fig. 1.

We conduct extensive experiments to validate our proposed methods. The results demonstrate that both `FedPOB` and `FedPOB-Pref` achieve considerably better performance than the previous baseline methods in various tasks. Furthermore, we empirically verify that the performance of our al-

gorithms consistently improves as the number of participating agents increases, highlighting the benefits of our collaborative approach. In summary, our key contributions are as follows:

- We propose `FedPOB`, a novel algorithm for sample-efficient federated prompt optimization that enables multiple agents to collaborate on finding the best prompts without sharing their raw data.
- We extend our algorithm to the practical setting of preference-based feedback by introducing the `FedPOB-Pref` algorithm, which is based on federated linear dueling bandits.
- We conduct extensive experiments to validate our approach, demonstrating that our algorithms significantly outperform existing baselines and scale effectively with more agents.

## 2 PROBLEM SETTING

**Prompt Optimization.** We address the problem of black-box prompt optimization, where the objective is to find an optimal prompt $p$ that maximizes the performance of a black-box LLM on a given task $\mathbb{D} = (\mathbb{X}, \mathbb{Y})$. The task consists of a set of queries $\mathbb{X} = \{x_k\}$ and their corresponding ground-truth answers $\mathbb{Y} = \{y_k\}$. Since the internal parameters of the black-box LLMs (e.g., GPT-4o-mini) are inaccessible and only API queries are allowed, we model the performance of the LLM via an external score function. Specifically, we define

$$s(p \mid \mathbb{D}) = \mathbb{E}_{(x,y)\in\mathbb{D}}\Big[m(\text{LLM}(p,x),y)\Big], \tag{1}$$

in which $m$ is a metric function that compares the model response $\text{LLM}(p,x)$ induced by the prompt $p$ with the ground-truth answer $y$ and provides a score $s(p \mid \mathbb{D})$. The optimization target is then formulated as

$$p^* = \arg\max_{p\in\mathbb{P}} s(p \mid \mathbb{D}), \tag{2}$$

where $\mathbb{P}$ denotes the space of all possible prompts.

**Federated Prompt Optimization.** We extend the black-box prompt optimization problem to the federated setting, which involves multiple agents. We consider a scenario with a set of $N > 1$ agents, denoted by $\mathbb{A}$, who all aim to solve the same task $\mathbb{D}$. To account for agent heterogeneity, we allow each agent $a \in \mathbb{A}$ to have its own prompt space denoted as $\mathbb{P}_a$. This increases the generality of our setting by allowing each user to define a prompt space uniquely suited to their own preferences. Furthermore, each agent can generate its local prompt space $\mathbb{P}_a$ using existing techniques (Zhou et al., 2023) and does not need to share its local prompt space with other agents. This allows every agent to keep its own local set of prompts private and hence aligns well with the federated setting. As a result, the federated prompt optimization problem can be expressed as follows:

$$p_a^* = \arg\max_{p_a\in\mathbb{P}_a} \mathbb{E}_{(x,y)\in\mathbb{D}}\Big[m(\text{LLM}(p_a,x),y)\Big], \ \forall a \in \mathbb{A} \tag{3}$$

Here, each agent $a \in \mathbb{A}$ aims to find the optimal prompt $p_a^*$ from its own prompt space $\mathbb{P}_a$ that maximizes its performance on the task $\mathbb{D}$. To achieve greater sample efficiency, all agents in $\mathbb{A}$ collaborate without sharing their raw data (i.e., the history of tested prompts and their scores). This problem formulation naturally aligns with common paradigms in the federated bandit literature (Wang et al., 2020; Dai et al., 2023). Therefore, we adopt the federated bandit framework to tackle this problem.

**Feedback Model.** To solve the federated black-box prompt optimization problem, we cast the optimization process into an iterative protocol, where we sequentially select candidate prompts for evaluation. At each round $t$, each agent $a$ selects one or two candidate prompts and receives feedback. The selection of the prompts is guided by theoretically principled bandit policies, which leverage the collective observation history from all agents to achieve sample-efficient optimization (more details in Sec. 3). Depending on the type of feedback available, we consider two settings:

- **Score feedback:** In this setting, each agent selects a single prompt $p_{t,a}$ at each round $t$, and receives a numeric score $\hat{s}_{t,a}$ as feedback, which directly reflects the performance of the prompt $p_{t,a}$ on task $\mathbb{D}$. Specifically, given a validation set $\mathbb{D}_\mathbb{V}$ representing the task $\mathbb{D}$, the score can be obtained as follows: $\hat{s}_{t,a} = \mathbb{E}_{(x,y)\in\mathbb{D}_\mathbb{V}}\Big[m(\text{LLM}(p_{t,a},x),y)\Big]$.
- **Preference feedback:** In this setting, every agent $a$ selects a pair of prompts $(p_{t,a}^1, p_{t,a}^2)$ at round $t$, and observes a binary signal indicating which of the two performs better, i.e., which prompt yielded the better response. For example, such feedback may be directly provided by human

---

**Algorithm 1** FedPOB (Agent $a \in \mathbb{A}$)

---

1: **Initialize:** $W_{\text{sync}} = W_{\text{new,a}} = \mathbf{0}_{d \times d}$, $V_{t,a} = \lambda I_{d \times d}$, $b_{\text{sync}} = b_{\text{new,a}} = \mathbf{0}_d$, $t_{\text{last}} = 0$
2: **for** $t = 1, 2, \ldots, T$ **do**
3:     Compute $V_{t,a} \leftarrow \lambda I + W_{\text{sync}} + W_{\text{new},a}$
4:     Update local model $\hat{\theta}_{t,a} \leftarrow V_{t,a}^{-1}(b_{\text{sync}} + b_{\text{new},a})$
5:     Select prompt $p_{t,a} \leftarrow \arg\max_{p \in \mathbb{P}_a} \langle \hat{\theta}_{t,a}, u(p) \rangle + \nu ||u(p)||_{V_{t,a}^{-1}}$
6:     Query $p_{t,a}$ to observe score feedback $\hat{s}_{t,a}$
7:     Update $W_{\text{new},a} \leftarrow W_{\text{new},a} + u_{t,a} u_{t,a}^{\top}$, $b_{\text{new},a} \leftarrow b_{\text{new},a} + u_{t,a} \hat{s}_{t,a}$
8:     **if** $(t - t_{\text{last}}) \cdot \log(\det V_{t,a} / \det V_{\text{last},a}) > D$ **then**
9:         Send a communication request to the central server
10:    **if** a communication round is started **then**
11:       Upload $\{W_{\text{new},a}, b_{\text{new},a}\}$ to the central server. Reset $W_{\text{new,a}} = \mathbf{0}_{d \times d}, b_{\text{new,a}} = \mathbf{0}_d$
12:       Receive $\{W_{\text{sync}}, b_{\text{sync}}\}$ from server

---

**Algorithm 2** FedPOB (Central Server)

---

1: **if** Central server receives a communication request from *any agent* **then**
2:     Initiate a communication round
3: **receive** $\{ W_{\text{new},a}$ and $b_{\text{new},a} \}_{a \in \mathbb{A}}$ from each agent
4: Update $W_{\text{sync}} \leftarrow W_{\text{sync}} + \sum_{a \in \mathbb{A}} W_{\text{new},a}$ , $b_{\text{sync}} \leftarrow b_{\text{sync}} + \sum_{a \in \mathbb{A}} b_{\text{new},a}$
5: Broadcast $W_{\text{sync}}$ and $b_{\text{sync}}$ to all agents

---

evaluators (Lin et al., 2024a). Following the common practice from dueling bandits (Bengs et al., 2022), we assume that the preference feedback is generated by the Bradley–Terry–Luce (BTL) model (Hunter, 2004).

## 3 FEDERATED PROMPT OPTIMIZATION VIA BANDITS

We adopt *linear models*, rather than more complex ones such as neural networks, to learn the unknown reward function for federated prompt optimization. Accordingly, our FedPOB and FedPOB-Pref algorithms (illustrated in Fig. 1) are based on linear bandits (Abbasi-Yadkori et al., 2011) and linear dueling bandits (Bengs et al., 2022), respectively. This choice is motivated by the balance linear models offer between expressiveness, simplicity, and theoretical guarantees: (1) Modern text embedding techniques powered by transformers are sufficiently mature and effective (Shi et al., 2024; Hu et al., 2024), enabling a simple linear function to model the relationship between prompts and scores. (2) Linear models enable lightweight algorithmic designs. (3) Unlike federated neural bandits using neural networks for reward estimation (Dai et al., 2023), federated linear bandit methods provide theoretical guarantees on collaboration which ensure that *the performance improves as more agents join the federation* (Wang et al., 2020).

### 3.1 THE FEDPOB ALGORITHM: SCORE FEEDBACK

Following recent works on black-box prompt optimization (Shi et al., 2024; Hu et al., 2024), we first map each discrete prompt $p$ into a continuous embedding vector $u(p) \in \mathbb{U}$ using a pre-trained model. This allows us to leverage rich semantic representations and simplifies the optimization problem. We then model the score of a prompt for each agent $a$ using a linear model: $s_a = \langle \theta_a, u(p_a) \rangle$, which is standard in the multi-armed bandit literature (Abbasi-Yadkori et al., 2011).

**Local Prompt Selection.** At the beginning of each round $t$, in lines 3-4 of Algo. 1, each agent $a$ first updates its information matrix $V_{t,a}$ and estimated linear parameters $\hat{\theta}_{t,a}$ using (1) *the aggregated information from all agents* received from the central server (i.e., $W_{\text{sync}}$ and $b_{\text{sync}}$, more details below) and (2) its newly collected local information (i.e., $W_{\text{new},a}$ and $b_{\text{new},a}$). Next, using the parameters $V_{t,a}$ and $\hat{\theta}_{t,a}$, agent $a$ selects the next prompt to query following the Upper Confidence Bound (UCB) strategy (line 5 of Algo. 1):

$$p_{t,a} = \arg\max_{p \in \mathbb{P}_a} \langle \hat{\theta}_t, u(p) \rangle + \nu ||u(p)||_{V_{t,a}^{-1}} \tag{4}$$

---

**Algorithm 3** FedPOB-Pref (Agent $a \in \mathbb{A}$)

---

1: **Initialize:** $W_{\text{sync}} = W_{\text{new,a}} = \mathbf{0}_{d \times d}$, $\hat{\theta}_0 \sim \mathcal{N}(\mathbf{0}, \sigma^2 I_d)$ with small $\sigma^2$,
2: **for** $t = 1, 2, \ldots, T$ **do**
3:      Select first prompt $p_{t,a}^1 \leftarrow \arg\max_{p \in \mathbb{P}_a} \langle \hat{\theta}_{t-1}, u(p) \rangle$
4:      Select second prompt $p_{t,a}^2 \leftarrow \arg\max_{p \in \mathbb{P}_a} \langle \hat{\theta}_{t-1}, u(p) - u(p_{t,a}^1) \rangle + \beta_t \|u(p) - u(p_{t,a}^1)\|_{W_{\text{sync}}^{-1}}$
5:      Query $p_{t,a}^1, p_{t,a}^2$ to observe preference feedback $\hat{\omega}_{t,a} = \mathbb{1}(p_{t,a}^1 \succ p_{t,a}^2)$
6:      Update local model $\hat{\theta}_{t,a} \leftarrow \arg\min_{p \in \mathbb{P}_a} L_{t,a}(\theta) - \langle \nabla L_a(\hat{\theta}_{t-1,a}), \theta \rangle + \frac{\lambda}{2}\|\theta - \hat{\theta}_{t-1}\|^2$
7:      Update $\nabla L_a(\theta_{t,a}) \leftarrow \nabla L_a(\theta_{t-1,a}) - \lambda(\hat{\theta}_{t,a} - \theta_{t-1})$
8:      Compute $W_{\text{new,a}} = [u(p_{t,a}^1) - u(p_{t,a}^2)][u(p_{t,a}^1) - u(p_{t,a}^2)]^\top$
9:      Upload $\{\hat{\theta}_{t,a}, \nabla L_a(\hat{\theta}_{t,a}), W_{\text{new,a}}\}$ to server

---

Here the parameter $\nu$ balances *exploitation* (choosing prompts with large predicted rewards) and *exploration* (choosing prompts with large uncertainty). Next, we test the selected prompt $p_{t,a}$ using the validation set $\mathbb{D}_\mathbb{V}$, to obtain score feedback $\hat{s}_{t,a}$ (line 6 of Algo. 1). Then, we update the newly collected local information $W_{\text{new},a}$ and $b_{\text{new},a}$ (line 7 of Algo. 1).

**Agent-Server Communication.** To reduce the communication cost, we only start a communication round when the new information collected by any agent exceeds a threshold $D$, i.e., when the criterion in line 8 of Algo. 1 is satisfied. If a communication request is sent by any agent, the trusted central server initiates a communication round (line 1-2 of Algo. 2) and all agents upload their local parameters $W_{\text{new},a}$ and $b_{\text{new},a}$ to the central server (lines 10-11). The central server then aggregates these local parameters to produce synchronized parameters $W_{\text{sync}}$ and $b_{\text{sync}}$ (line 3-4 of Algo. 2), which are then broadcast to all agents. After the agents receive the aggregated parameters $W_{\text{sync}}$ and $b_{\text{sync}}$, they can use them to select the prompt in the next iteration, and the algorithm repeats.

### 3.2 THE FEDPOB-PREF ALGORITHM: PREFERENCE FEEDBACK

In many practical applications, obtaining explicit numerical scores is challenging, whereas collecting pairwise preference feedback is often more natural and cost-effective. Specifically, in human-in-the-loop scenarios, users can more reliably state a preference between two generated outputs than assign them absolute scores (Yue et al., 2012; Lin et al., 2024a). For example, in LLM-based creative writing, it is often more intuitive for users to express a preference between two generated articles than to quantify subjective alignment with a numeric score. Similarly, in text-to-image generation, it is often more practical to rely on pairwise comparisons than to capture complex aesthetic preferences with a single numeric score. This setting, however, introduces a significant technical hurdle: *the parameter estimation for linear dueling bandits does not have a closed-form solution* (Bengs et al., 2022). This limitation prevents the use of the simple parameter aggregation strategy employed by our FedPOB algorithm.

The absence of a closed-form solution naturally leads to gradient-based optimization approaches. Recent work by Huang et al. (2025) introduced federated linear dueling bandit algorithms (FLDB-GD and FLDB-OGD) that achieve collaboration by aggregating local gradients. While theoretically sound, these methods face a practical dilemma: FLDB-GD incurs high communication costs, whereas the more communication-efficient FLDB-OGD suffers significant performance degradation. We attribute this to the fact that *preference feedback is inherently noisier and less informative than numerical scores*, making it particularly challenging to achieve both competitive performance and communication efficiency. To overcome this, we draw inspiration from *classical federated learning* for solving supervised learning problems (McMahan et al., 2017). Specifically, instead of aggregating gradients, we aggregate model parameters, which allows us to adopt a dynamic regularization technique that has proven effective in federated learning (Acar et al., 2021) for further performance improvement. This leads to our proposed FedPOB-Pref algorithm (Algos. 3 and 4).

Our FedPOB-Pref algorithm offers several key advantages: (1) it is highly **sample-efficient**, capable of learning the underlying reward model from a small number of preference queries; (2) it is robust to **agent heterogeneity**, and its performance scales effectively with the number of collaborating agents; and (3) when compared to the baselines from Huang et al. (2025), FedPOB-Pref simultaneously **reduces communication costs and improves performance** (Sec. 4.2).

---

**Algorithm 4** `FedPOB-Pref` (Central Server)

1: **receive** $\{\hat{\theta}_{t,a}, \nabla L_a(\hat{\theta}_{t,a}), W_{\text{new,a}}\}_{a \in \mathbb{A}}$ from each agent
2: Update server model $\hat{\theta}_t \leftarrow \frac{1}{n}\sum_{a \in \mathbb{A}} \hat{\theta}_{t,a} - \frac{1}{n}\sum_{a \in \mathbb{A}} \frac{1}{\lambda}\nabla L_a(\hat{\theta}_{t,a})$
3: Update $W_{\text{sync}} \leftarrow W_{\text{sync}} + \sum_{a \in \mathbb{A}} W_{\text{new},a}$
4: Broadcast $\hat{\theta}_t$ and $W_{\text{sync}}$ to all agents

---

The overall workflow of `FedPOB-Pref` is outlined in Algorithms 3 and 4. At each round $t$, every agent $a$ selects a pair of prompts based on the global model $\hat{\theta}_{t-1}$. The first prompt, $p_{t,a}^1$, represents pure **exploitation** (line 3), while the second, $p_{t,a}^2$, incorporates an **exploration** bonus to discover more informative options (line 4). This dueling selection strategy is grounded in the theory of dueling bandits (Bengs et al., 2022; Verma et al., 2024). We then obtain binary preference feedback $\omega_{t,a} = \mathbb{1}_{p_{t,a}^1 \succ p_{t,a}^2}$ for this pair of selected prompts (line 5). The core of our method lies in the local model update (line 6), which optimizes an objective that combines the standard logistic loss with a dynamic regularizer (Acar et al., 2021). The first component is the pairwise logistic loss over the agent's local history:

$$L_{t,a}(\theta) = -\sum_{\tau=1}^{t-1} \Big( \omega_{\tau,a} \log \sigma\big(\theta^\top [u(p_{\tau,a}^1) - u(p_{\tau,a}^2)]\big) + (1-\omega_{\tau,a}) \log \sigma\big(\theta^\top [u(p_{\tau,a}^2) - u(p_{\tau,a}^1)]\big) \Big).$$
(5)

This term is the negative log-likelihood of the observed preferences under the BTL model (Bengs et al., 2022). The second component is a dynamic regularization term consisting of (i) a linear penalty, $-\langle \nabla L_a(\hat{\theta}_{t-1,a}), \theta \rangle$, which corrects for local gradient drift, and (ii) a quadratic penalty, which prevents the local model from deviating excessively from the previous global model (Acar et al., 2021). After this local update (lines 6-8), agents upload their new parameters to the central server for aggregation, which then broadcasts the aggregated global parameters for the next round. Of note, we conduct theoretical analysis to motivate the local objective function of `FedPOB-Pref` (App. E), providing theoretical justification for its strong performance (Sec. 4.2).

## 4 EXPERIMENTS

We adopt MPNet (Song et al., 2020) as the text embedding model, and use GPT-3.5-turbo (OpenAI, 2023a) in the experiments unless specified otherwise. Of note, we also test two other models, GPT-4o-mini (OpenAI, 2023b) and Qwen3-235B-A22B-2507 (Bai et al., 2023), in Sec. 5. Evaluation is performed on the Instruction Induction (Chen et al., 2023; Lin et al., 2024b) and BIG-Bench Hard datasets (Suzgun et al., 2023), which collectively cover over 50 tasks that span diverse areas such as reasoning, language comprehension, and code generation. To account for agent heterogeneity, we ensure that the prompt domains of all agents contain both shared prompts and unique prompts. For fair comparisons, we ensure an equal validation query budget across all algorithms and analyze the corresponding communication costs in the federated setting. We defer more details on the experimental setting to App. C.

### 4.1 SCORE FEEDBACK: FEDPOB

In the setting with score-based feedback, every tested prompt receives a numerical score indicating the quality of its induced response. Here we assess performance of a prompt using a validation set and adopt the validation accuracy as the corresponding score. The objective is to identify the optimal prompt (i.e., the one that achieves the highest validation score). We compare our `FedPOB` with a representative baseline method on federated prompt optimization: FedOne (Wang et al., 2025), as well as two other baselines on standard prompt optimization: INSTINCT (Lin et al., 2024b) and PromptBreeder (Fernando et al., 2024).

Table 1 and 2 report the final scores achieved by the best prompt discovered by each algorithm in various tasks. The results demonstrate the superior capability of our `FedPOB`, which achieves the highest score on the majority of the tasks under the setting of ten agents. The results also show the sample efficiency of our `FedPOB` since it achieves the best performance given a fixed number of samples per agent. Fig. 2 depicts the performance of `FedPOB` across different iterations,

Table 1: Average validation accuracy (with standard error) of the best prompt found by each algorithm in the **Instruction Induction dataset**, averaged over 5 independent trials with different random seeds. For clarity, only a representative subset of challenging tasks. The complete results for all tasks are provided in Table 5 (App. D.3) and the results are consistent.

| Dataset | INSTINCT | PromptBreeder | FedOne (10 agents) | FedPOB (ours) | | |
|---|---|---|---|---|---|---|
| | | | | 1 Agent | 3 Agents | 10 Agents |
| Active to Passive | 0.940±0.053 | **1.000±0.000** | **1.000±0.000** | 0.804±0.160 | 0.960±0.014 | 0.972±0.023 |
| Auto Categorization | **0.313±0.012** | 0.220±0.020 | 0.264±0.004 | 0.272±0.030 | 0.308±0.018 | 0.288±0.023 |
| Antonyms | 0.767±0.023 | 0.840±0.020 | **0.870±0.005** | 0.792±0.046 | 0.812±0.027 | 0.828±0.023 |
| Common Concept | **0.217±0.040** | 0.118±0.010 | 0.136±0.003 | 0.188±0.015 | 0.210±0.007 | 0.208±0.018 |
| Informal to Formal | 0.570±0.020 | 0.521±0.067 | **0.605±0.005** | 0.528±0.028 | 0.528±0.039 | 0.570±0.030 |
| Larger Animal | **0.993±0.012** | 0.987±0.012 | 0.829±0.037 | 0.984±0.017 | 0.992±0.011 | 0.989±0.011 |
| Negation | 0.860±0.020 | 0.927±0.012 | 0.897±0.010 | 0.856±0.061 | **0.940±0.014** | 0.920±0.032 |
| Orthography Starts With | 0.767±0.214 | 0.813±0.061 | 0.436±0.024 | 0.804±0.100 | 0.828±0.056 | **0.832±0.087** |
| Rhymes | 0.493±0.142 | 0.393±0.031 | 0.916±0.027 | 0.664±0.120 | 0.776±0.187 | **0.844±0.106** |
| Second Word Letter | 0.847±0.110 | 0.947±0.042 | 0.625±0.034 | 0.792±0.199 | 0.880±0.157 | **0.972±0.023** |
| Sentence Similarity | 0.467±0.031 | 0.380±0.020 | 0.360±0.035 | **0.540±0.094** | 0.508±0.082 | 0.448±0.018 |
| Sentiment | 0.973±0.012 | 0.993±0.012 | **0.996±0.002** | 0.988±0.018 | 0.972±0.023 | 0.972±0.027 |
| Synonyms | 0.327±0.150 | 0.333±0.115 | 0.320±0.023 | 0.324±0.103 | 0.296±0.041 | **0.384±0.124** |
| Taxonomy Animal | 0.947±0.023 | 0.967±0.042 | 0.805±0.026 | 0.924±0.073 | **0.980±0.024** | 0.972±0.034 |
| Translation En-De | 0.820±0.020 | 0.820±0.060 | **0.927±0.004** | 0.820±0.047 | 0.840±0.032 | 0.868±0.036 |
| Translation En-Es | 0.747±0.042 | 0.746±0.023 | **0.950±0.012** | 0.756±0.026 | 0.740±0.072 | 0.728±0.030 |
| Translation En-Fr | 0.947±0.023 | 0.920±0.040 | 0.919±0.005 | 0.944±0.033 | 0.940±0.283 | **0.948±0.018** |
| Word in Context | 0.553±0.058 | 0.620±0.040 | 0.409±0.091 | 0.460±0.084 | **0.640±0.020** | 0.608±0.036 |
| Object Counting | 0.520±0.106 | 0.473±0.110 | 0.497±0.019 | 0.520±0.074 | **0.616±0.039** | 0.588±0.050 |
| Odd One Out | 0.867±0.058 | 0.833±0.116 | 0.859±0.024 | 0.800±0.122 | **0.900±0.000** | **0.900±0.000** |
| Word Sorting | 0.753±0.058 | 0.753±0.099 | 0.497±0.026 | 0.756±0.093 | 0.744±0.065 | **0.828±0.063** |
| Word Unscrambling | 0.687±0.012 | 0.687±0.023 | **0.728±0.005** | 0.724±0.046 | 0.716±0.026 | 0.720±0.028 |
| Average (22 Tasks) | 0.669 | 0.665 | 0.645 | 0.663 | 0.701 | **0.712** |

Table 2: Performance on the **Big-Bench Hard (BBH) dataset** under the same experimental settings.

| Dataset | INSTINCT | PromptBreeder | FedOne (10 agents) | FedPOB (ours) | | |
|---|---|---|---|---|---|---|
| | | | | 1 Agent | 3 Agents | 10 Agents |
| Boolean Expressions | 0.793±0.046 | 0.853±0.012 | **0.883±0.003** | 0.800±0.025 | 0.836±0.021 | 0.844±0.026 |
| Date Understanding | 0.587±0.012 | 0.593±0.030 | **0.633±0.007** | 0.580±0.028 | 0.576±0.033 | 0.572±0.030 |
| Disambiguation QA | 0.713±0.031 | 0.753±0.023 | **0.858±0.011** | 0.816±0.026 | 0.844±0.017 | 0.840±0.032 |
| Dyck Languages | 0.713±0.031 | 0.693±0.012 | **0.722±0.005** | 0.672±0.018 | 0.668±0.023 | 0.680±0.032 |
| Formal Fallacies | 0.687±0.031 | 0.967±0.058 | **0.991±0.002** | 0.700±0.121 | 0.872±0.175 | 0.812±0.172 |
| Geometric Shapes | 0.453±0.058 | 0.360±0.060 | 0.272±0.007 | 0.436±0.022 | 0.412±0.039 | **0.448±0.036** |
| Hyperbaton | 0.913±0.046 | 0.907±0.023 | 0.946±0.003 | 0.868±0.522 | 0.928±0.027 | **0.948±0.018** |
| Logical Deduction Five Objects | 0.473±0.046 | 0.460±0.053 | 0.466±0.009 | 0.464±0.041 | 0.452±0.030 | **0.476±0.017** |
| Logical Deduction Seven Objects | 0.513±0.046 | 0.473±0.031 | 0.485±0.002 | 0.476±0.043 | **0.492±0.046** | 0.488±0.415 |
| Logical Deduction Three Objects | 0.600±0.053 | 0.573±0.046 | 0.635±0.009 | 0.604±0.033 | 0.636±0.017 | **0.644±0.009** |
| Movie Recommendation | **0.820±0.069** | 0.767±0.023 | 0.688±0.004 | 0.720±0.037 | 0.720±0.032 | 0.732±0.027 |
| Multistep Arithmetic Two | 0.647±0.129 | 0.601±0.030 | 0.685±0.017 | 0.580±0.105 | 0.648±0.018 | **0.692±0.046** |
| Navigate | 0.707±0.031 | 0.760±0.020 | 0.755±0.028 | 0.688±0.052 | 0.720±0.042 | **0.716±0.026** |
| Penguins in a Table | 0.577±0.031 | **0.694±0.016** | 0.581±0.031 | 0.562±0.035 | 0.584±0.031 | 0.605±0.015 |
| Reasoning about Colored Objects | 0.547±0.023 | 0.593±0.023 | 0.440±0.008 | 0.548±0.036 | 0.528±0.034 | **0.568±0.027** |
| Ruin Names | 0.707±0.023 | **0.767±0.042** | 0.625±0.003 | 0.688±0.039 | 0.660±0.042 | 0.724±0.067 |
| Salient Translation Error Detection | 0.573±0.012 | **0.633±0.070** | 0.500±0.055 | 0.584±0.033 | 0.588±0.018 | 0.600±0.028 |
| Snarks | 0.778±0.022 | 0.770±0.051 | 0.675±0.003 | 0.779±0.022 | **0.791±0.012** | 0.782±0.019 |
| Sports Understanding | 0.440±0.106 | 0.540±0.072 | **0.669±0.004** | 0.524±0.114 | 0.552±0.073 | 0.564±0.078 |
| Temporal Sequences | 0.647±0.050 | 0.473±0.046 | 0.403±0.019 | 0.612±0.058 | 0.648±0.050 | **0.652±0.052** |
| Tracking Shuffled Objects Five Objects | 0.300±0.053 | 0.287±0.012 | 0.279±0.030 | 0.296±0.017 | 0.304±0.017 | **0.328±0.023** |
| Tracking Shuffled Objects Seven Objects | 0.280±0.020 | 0.253±0.042 | **0.281±0.006** | 0.268±0.023 | 0.268±0.023 | 0.256±0.029 |
| Tracking Shuffled Objects Three Objects | **0.473±0.046** | 0.440±0.020 | 0.413±0.018 | 0.432±0.039 | 0.420±0.049 | 0.400±0.014 |
| Web of Lies | 0.633±0.023 | 0.607±0.012 | 0.627±0.012 | 0.640±0.039 | **0.644±0.043** | 0.636±0.026 |
| Average (24 Tasks) | 0.607 | 0.618 | 0.605 | 0.596 | 0.616 | **0.625** |

where we observe a positive correlation between the number of agents and the achieved prompt score, highlighting the benefits of multi-agent collaboration and improved sample efficiency with more agents. In addition, FedPOB achieves a near-optimal score with a small batch of samples, demonstrating its sample efficiency.

## 4.2 PREFERENCE FEEDBACK: FedPOB-Pref

To simulate user preference feedback in our experiments, we adopt the protocol from Lin et al. (2024a). For any pair of prompts $(p_{t,1}, p_{t,2})$, we first compute their ground-truth scores, $s(p_{t,1})$ and $s(p_{t,2})$, on a validation set. The preference probability is then determined by the Bradley-

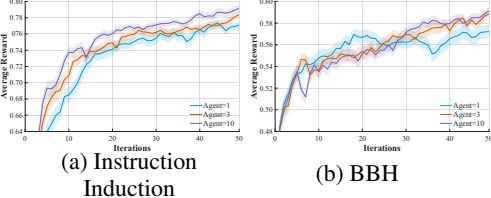

(a) Instruction Induction  (b) BBH

Figure 2: Performance of `FedPOB` with varying numbers of agents.

(a) Instruction Induction  (b) BBH

Figure 3: Performance of `FedPOB-Pref` with varying numbers of agents.

Terry-Luce (BTL) model (Hunter, 2004): $P(p_{t,1} \succ p_{t,2}) = \sigma(s(p_{t,1}) - s(p_{t,2}))$, where $\sigma(\cdot)$ is the sigmoid function. A binary preference outcome $y_t = \mathbb{1}(p_{t,1} \succ p_{t,2})$ is then sampled from a Bernoulli distribution with this probability. We compare `FedPOB-Pref` against federated baselines FLDB-GD and FLDB-OGD (Huang et al., 2025), as well as standard prompt optimization methods APOHF (Lin et al., 2024a) and DoubleTS (Dwaracherla et al., 2024).

The results, summarized in Table 3, demonstrate that given the same number of samples per agent, `FedPOB-Pref` consistently achieves the best performance across different numbers of agents. This showcases the superior sample efficiency of our `FedPOB-Pref`. Our method establishes a superior trade-off between performance and communication cost. Specifically, `FedPOB-Pref` matches the communication efficiency of FLDB-OGD while delivering substantially better results. Conversely, while FLDB-GD obtains the second-best performance, it does so at a considerably higher communication cost. Fig. 3 further highlights that the sample efficiency of `FedPOB-Pref` improves as more agents collaborate. Additional results are available in Fig. 10 (App. D.2).

Table 3: Score and number of communication rounds under **preference feedback**.

| Method | Agent | Instruction Induction | | BBH | |
|---|---|---|---|---|---|
| | | Perf. | Comm. | Perf. | Comm. |
| APOHF | - | 0.7681 | - | 0.5838 | - |
| Double TS | - | 0.7859 | - | 0.5983 | - |
| FLDB-GD | 1 | 0.7624 | 1500 | 0.5868 | 1500 |
| | 3 | 0.7959 | 1500 | 0.6204 | 1500 |
| | 10 | 0.8244 | 1500 | 0.6457 | 1500 |
| FLDB-OGD | 1 | 0.6872 | **50** | 0.5286 | **50** |
| | 3 | 0.7687 | **50** | 0.5880 | **50** |
| | 10 | 0.8123 | **50** | 0.6271 | **50** |
| FedPOB-Pref | 1 | **0.8000** | 50 | **0.6213** | 50 |
| | 3 | **0.8145** | 50 | **0.6357** | 50 |
| | 10 | **0.8482** | 50 | **0.6583** | 50 |

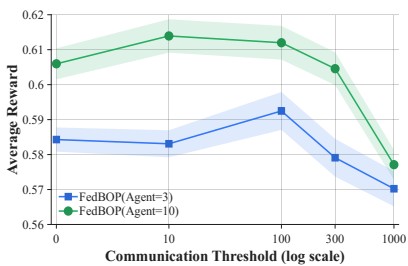

Figure 4: Scores of `FedPOB` with varying communication thresholds $D$.

## 5 ABLATION STUDY

**Performance vs. Communication in `FedPOB`.** In federated learning, communication is inherently costly, making frequent interactions with the central server impractical. Thus, an effective algorithm should maintain strong performance even with infrequent communications. Here we reduce the interaction frequency by varying the communication threshold $D$ in `FedPOB` in the range: $\{0, 10, 100, 300, 1000\}$. Note that a larger $D$ results in less communication rounds, and we report the number of communication rounds in App. G.3. The results in Fig. 4 reveal a clear trade-off between performance and communication, i.e., fewer communication rounds (i.e., larger $D$) result in worse performance. More importantly, our `FedPOB` still achieves strong performance even with infrequent communications, demonstrating its robustness and practical effectiveness in realistic federated environments.

**Generalization to Other LLMs.** While the response quality of an LLM depends not only on the prompt design but also on the inherent capability of the backbone model, we examine whether the observed performance gains of our algorithms can generalize to other LLMs. To this end,

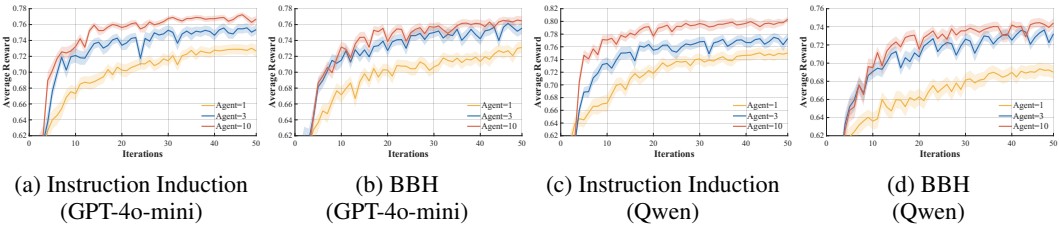

| (a) Instruction Induction (GPT-4o-mini) | (b) BBH (GPT-4o-mini) | (c) Instruction Induction (Qwen) | (d) BBH (Qwen) |

Figure 5: The performance of `FedPOB` using GPT-4o-mini and Qwen.

we replace the GPT-3.5-Turbo model used in our main experiments by GPT-4o-mini and Qwen (OpenAI, 2023a;b; Bai et al., 2023), while keeping all other settings fixed. As shown in Fig. 5, our `FedPOB` consistently discovers high-score prompts and achieves better performance with a larger number of agents, regardless of the underlying LLM. Additional results on the performance of `FedPOB-Pref` can be found in App. D.4, which lead to consistent observations.

**Effectiveness of Dynamic Regularization in `FedPOB-Pref`.** We further assess the necessity of the dynamic regularization term in `FedPOB-Pref`, which mitigates the dynamic drift among heterogeneous clients and accelerates collaboration. We compare the performance of `FedPOB-Pref` with and without this term, the latter of which is equivalent to the classical FedAvg algorithm (McMahan et al., 2017)). Fig. 6 shows that incorporating dynamic regularization stabilizes performance, speeds up convergence, and reduces fluctuations caused by inter-agent heterogeneity. These results highlight its critical role in enabling efficient and robust federated prompt optimization in heterogeneous federated environments.

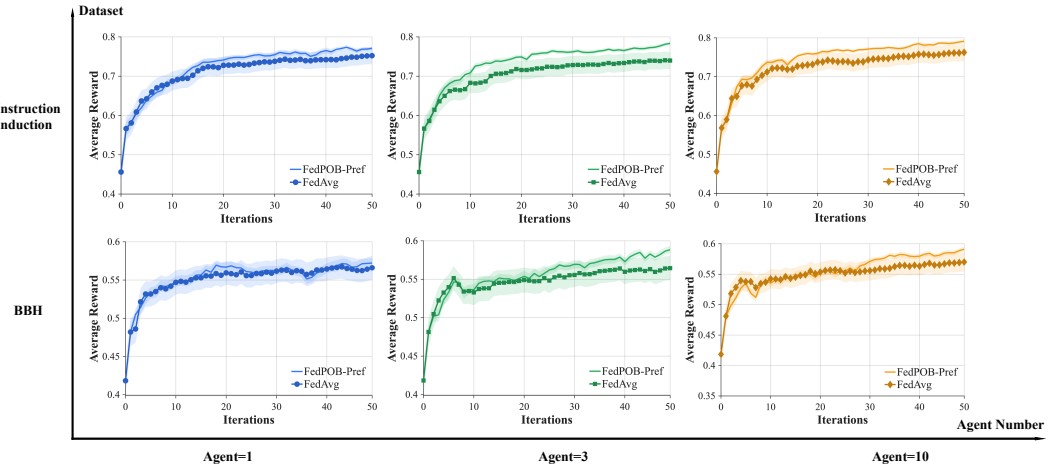

Figure 6: Impact of the dynamic regularization term in `FedPOB-Pref`. FedAvg corresponds to removing this term.

## 6 RELATED WORK

**Federated Prompt Optimization.** Federated Learning enables collaborative model training without sharing private data (Kairouz et al., 2019; McMahan et al., 2017). However, applying FL to LLMs faces a critical barrier: the prohibitive cost of communicating updates for models of such massive scale. A natural workaround is to combine FL with parameter-efficient prompt tuning (Zhao et al., 2023; Che et al., 2023; Deng et al., 2024; Wei et al., 2023), where only lightweight soft prompts are trained and communicated. While resource-efficient, this paradigm operates in a white-box setting and thus fails in API-based black-box scenarios. This limitation has motivated research on black-box federated prompt optimization (Lin et al., 2023a). Early efforts such as FedBPT (Zhang et al., 2023) adopt soft prompts with gradient-free optimization, but remain incompatible with API-only LLMs. More recent work addresses discrete prompt optimization, e.g., FedOne (Wang et al.,

2025), which learns categorical distributions to sample prompts. Despite solving discreteness, these methods suffer from inefficiency and poor semantic quality, leaving open the challenge of developing a query-efficient federated method that produces semantically meaningful discrete prompts for black-box LLMs. We defer a detailed discussion of the related works on standard non-federated prompt optimization to App. B due to space constraint.

## 7 CONCLUSION AND FUTURE WORK

In this paper, we introduced `FedPOB` and `FedPOB-Pref`, novel algorithms for sample-efficient federated prompt optimization. Built upon the theory of federated multi-armed bandits, our methods enable multiple agents to effectively collaborate to find optimal prompts for black-box LLMs without sharing raw data. Extensive experiments demonstrate that our algorithms significantly outperform existing baselines under both score and preference feedback, with performance consistently improving with an increasing number of participating agents. Notably, `FedPOB-Pref` establishes a superior performance-to-communication trade-off in the practical preference-based setting. A promising future direction is extending our algorithms to the asynchronous communication setting. In addition, our current study focuses on generative LLMs, and extending our framework to pure encoder tasks (e.g., RoBERTa on GLUE) remains another interesting direction for future work. Regarding `FedPOB-Pref`, promising future directions include establishing stronger theoretical guarantees and exploring alternative pairwise comparison models beyond the BTL framework.

## REPRODUCIBILITY STATEMENT

To ensure reproducibility, we have uploaded the code in the supplementary material. We have also clearly described all detailed experimental settings (Sec. 4 and App. C) to ensure transparency and reproducibility.

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

## A  LLM USAGE DISCLOSURE

We acknowledge the ICLR 2026 policies on LLM usage. All scientific ideas, proofs, experimental design, and conclusions are the original work of the human authors. LLMs were used solely as writing assistants to polish the text for minor grammar correction and clarification. All LLM-assisted text has been carefully reviewed by the authors.

## B  ADDITIONAL RELATED WORK

The performance of Large Language Models (LLMs) is highly sensitive to the quality of input prompts (Zhou et al., 2023; Lin et al., 2024b). While carefully handcrafted prompts (Brown et al., 2020; Wei et al., 2022) can substantially enhance model capabilities, the manual design process is time-consuming and heavily reliant on expert intuition. To address this challenge, early studies focused on white-box prompt optimization, including AutoPrompt (Shin et al., 2020), Prefix-Tuning (Li & Liang, 2021), P-Tuning (Liu et al., 2021), and Prompt Tuning (Lester et al., 2021). More recently, increasing attention has been devoted to black-box prompt optimization (Yang et al., 2024; Mañas et al., 2024; Juneja et al., 2025; Schneider et al., 2024), with representative methods such as GRIPS (Prasad et al., 2023), BDPL (Diao et al., 2023), PRewrite (Kong et al., 2024), PromptAgent (Wang et al., 2024), and APO (Pryzant et al., 2023). RLPrompt (Deng et al., 2022) addresses the discrete black-box setting by optimizing a probability distribution over prompts, from which candidates are sampled to identify the optimal one. Evolutionary approaches, such as EvoPrompt (Guo et al., 2024) and Promptbreeder (Fernando et al., 2024), employ mutation and crossover to iteratively improve prompts. Zhou et al. (Zhou et al., 2023) introduced APE, which leverages an LLM to generate candidate instructions and refines those with high evaluation scores. However, these approaches often require extensive sampling and validation, making them sample-inefficient. A key direction has been reframing black-box prompt optimization as a continuous problem, as in InstructZero (Chen et al., 2023) and ZOPO (Hu et al., 2024). Building on this idea, INSTINCT (Lin et al., 2024b) employs neural bandits to sequentially select instructions to query, leveraging neural networks to better capture the relationship between prompts and their performance, thereby enabling more efficient optimization.

Recent work has investigated prompt optimization in scenarios where direct human feedback is difficult to obtain and only preference feedback is available. BPO (Luo et al., 2023) trains an independent optimizer that automatically rewrites initial prompts using paired preference data, encouraging black-box LLMs to produce better responses. Align-Pro (Ye et al., 2023) develops a theoretical framework based on the Bradley–Terry model to analyze and guide optimization through pairwise comparisons. APOHF (Lin et al., 2024a) formulates prompt optimization as a dueling bandits problem, directly leveraging pairwise preferences (e.g., A is better than B) to efficiently identify the best prompt among candidates. Building on this idea, PLHF (Yang et al., 2025) extends preference-based optimization to a few-shot setting, demonstrating that high-quality prompts can be identified with only a small number of comparisons, thereby greatly reducing annotation costs. In addition, the prompt selection strategy of our `FedPOB-Pref` algorithm is also related to the best-arm-identification (BAI) version of the method from the work of Liu et al. (2024), which also aims to achieve exploration by encouraging the selection of a response with large uncertainty.

**Private Evolution.** A related line of work explores Private Evolution, which similarly utilizes scores to guide iterative queries to black-box LLMs (Lin et al., 2023b; Xie et al., 2024; Lin et al., 2025). However, these methods differ fundamentally from our approach in their privacy mechanisms and optimization frameworks. While the Private Evolution literature primarily leverages *Differential Privacy (DP)* to generate synthetic data—with recent extensions to federated settings relying on preference-optimized synthetic data (Hou et al., 2025)—our framework adopts a *Federated Learning* paradigm based on direct parameter aggregation without exposing raw data.

# C MORE DETAILS ON THE EXPERIMENTAL SETTING

## C.1 DATASETS AND MODELS

**Datasets.** We use 29 tasks from the Instruction-Induction dataset (Lin et al., 2024b), excluding the auto-debugging task which contains only 8 instances, and the Cause-and-Effect task. The Cause-and-Effect task is an open-ended reasoning problem where multiple answers may be reasonable, but only one ground-truth is provided. Existing metrics cannot accurately evaluate responses, and most automatic scores are generally zero. For example, a few instances are:

- Cause: "The child hurt their knee." Effect: "The child started crying."
- Cause: "My car got dirty." Effect: "I washed the car."
- Cause: "Someone fainted." Effect: "Someone called 911."

For the BBH dataset (Suzgun et al., 2023), we adopt 24 tasks, excluding 3 tasks that overlap with Instruction-Induction to avoid double evaluation.

**Models.** Our experiments are conducted on three LLMs, *OpenAI/GPT-3.5-turbo-0613*, *OpenAI/GPT-4o-mini*, and *Qwen/Qwen3-235B-A22B-2507* via the OpenRouter API. We use MP-Net (Song et al., 2020) as the embedding model.

## C.2 PROMPT SPACE GENERATION

To simulate a realistic federated setting, we adopt the APE algorithm (Zhou et al., 2023) to construct a prompt pool from a small initial task description (i.e., a set of input–output exemplars). From this pool, each agent samples both shared and personalized prompts, thereby capturing the inherent data heterogeneity—where shared prompts model the common knowledge across agents, while personalized prompts reflect the distinct distributions, preferences, and contextual variations specific to each client.

**Prompt Template.** We follow INSTINCT (Lin et al., 2024b) for prompt template to automatically generate prompt space. We use 5 exemplars in datasets to query LLM to induct prompt.

---

**Prompt Generation Template**

Input: [INPUT]
Output: [OUTPUT]
<More exemplars...>
Input: [INPUT]
Output: [OUTPUT]
The instruction was to

---

Figure 7: Prompt Generation template for prompt space generation.

---

**Prompt Generation Example**

Input: [Today is Christmas Eve of 1937. What is the date 10 days later?]
Output: [01/03/1938]
<More exemplars...>
Input: [Jane thought today is 3/11/2002, but today is in fact Mar 12, which is 1 day later. What is the date 24 hours later?]
Output: [03/13/2002]
The instruction was to

---

Figure 8: Illustrative example of prompt generation with the template.

### C.3   IMPROVED EVALUATION METHOD

**Evaluation Challenges.** Due to the complex nature of the BBH tasks, we observed that large language models (LLMs) often generate detailed explanations along with their final answers, unlike the more direct outputs seen in the Instruction-Induction tasks. This behavior was particularly prevalent when using models such as GPT-4o-mini and Qwen3. A small number of tasks in the Instruction-Induction dataset also exhibited this tendency toward verbose responses. Standard evaluation metrics such as exact match, contain, or F1-score proved unreliable in this context. Since the ground-truth answers are typically concise, the verbosity of model outputs frequently led to misclassification. In some cases, a model's response was fully correct from a human perspective, yet automated metrics incorrectly assigned a score of zero.

**Multi-choice Metric.** To mitigate this issue, we designed a new evaluation metric, termed Multi-choice, specifically tailored to handle the verbose outputs of LLMs on BBH tasks. Our approach normalizes the model's output and checks whether the ground-truth answer is present. In practice, we extract the final sentence of the model's prediction and verify if it contains the ground-truth answer.

**Metrics.** For BBH, we evaluate on 24 tasks using the Multi-choice metric. For Instruction-Induction (29 tasks), we follow Lin et al. (2024b) and adopt the same evaluation setup. Concretely, we use the F1 metric for "Common concept" and "Informal to formal"; exact set matching for "Orthography starts with" and "Taxonomy animal"; and label containment for "Synonyms". For the remaining tasks, we apply exact match. Additionally, for "Diff" and "Odd one out", when evaluated with GPT-4o-mini or Qwen3 (where verbose explanations are frequent), we employ the Multi-choice metric instead of exact match.

**Cached Prompt Scoring.** We leverage the alignment between prompts and their validation scores. Since our validation set is relatively large (50 samples), we observed that the scores obtained for a given prompt remain stable across repeated evaluations. Consequently, for all algorithms that require optimization over a prompt space (excluding FedOne and PromptBreeder, which do not depend on a prompt space), we evaluate each prompt once on the validation set and cache the resulting score for subsequent use. This strategy substantially reduces computation time while maintaining evaluation reliability.

### C.4   HYPERPARAMETERS OF OUR ALGORITHMS

In `FedPOB`, we set $\lambda = 1$, $\nu = 0.3$, $D = 10.0$, and $d = 768$, where $d$ matches the output feature dimension of MPNet (Song et al., 2020). For `FedPOB-Pref`, we set $\lambda = 1$ and use a learning rate of 0.001 to update $\theta_{t,a}$ (line 7 of Algo. 3). Training is conducted for 30 iterations.

The parameter $\beta_t$ is time-dependent. Following (Huang et al., 2025), we set

$$\beta_t = \sqrt{2\log(1/\delta) + d\log\left(1 + \frac{t\kappa_\mu}{d\lambda}\right)},$$

where $\kappa_\mu$ denotes the number of agents and $d$ is the feature dimension (here $d = 768$ for compatibility with MPNet).

### C.5   HYPERPARAMETERS OF BASELINE AND FAIR COMPARISONS

To ensure fairness, we set the total number of validation queries to be the same across all methods and report them consistently in our experimental results (see Tables 1, 2, and 3 in Sec. 4, as well as Table 5 in App. D).

For score feedback baselines, only INSTINCT and our method share the same evaluation protocol, where each iteration queries the validation set once. Therefore, we ensure fairness by comparing the best reward obtained within the first 50 validation queries, rather than rewards at every single iteration. For preference-feedback baselines, all methods query the validation set twice per iteration, as two prompts are sampled for pairwise comparison. Running 50 iterations thus corresponds to 100 validation queries in total. For consistency, we report the score of the **first** (exploitation) prompt selected by each method. This is consistent with the work of Lin et al. (2024a). The reward curves

are plotted across iterations, where the $x$-axis represents the number of iterations (equivalently, preference-feedback steps).

Table 4: Query settings and reported metrics for different methods.

| Score Feedback | | | |
|---|---|---|---|
| **Method** | **Queries/Iter** | **Total Queries** | **Reported Metric** |
| FedPOB | 1 | 50 | Best reward at 50th iter. |
| INSTINCT | 1 | 50 | Best reward at 50th iter. |
| PromptBreeder | 5 | 50 | Best reward at 10th round. |
| FedOne | 5 | 50 | Best reward at 50th iter. |
| **Preference Feedback** | | | |
| FedPOB-Pref | 2 | 100 | Best reward at 50th iter. |
| FLDB-OGD | 2 | 100 | Best reward at 50th iter. |
| FLDB-GD | 2 | 100 | Best reward at 50th iter. |
| APOHF | 2 | 100 | Best reward at 50th iter. |
| Double-TS | 2 | 100 | Best reward at 50th iter. |

**Score Feedback.** For FedPOB, we run 50 iterations, thus querying the validation set 50 times. We report the best reward at the 50th iteration. For INSTINCT, we follow the default settings from their paper, which are consistent with our protocol (one query per iteration), and also report the best reward at the 50th iteration. For PromptBreeder, which is an evolutionary algorithm, half of the population queries the validation set in each round. With a population size of 10 (2 mutation prompts $\times$ 5 thinking styles), this results in 5 queries per round and 50 queries in total over 10 rounds; we report the best reward at the 10th round. For FedOne, we follow the original paper and construct its vocabulary using the PMI algorithm, sampling frequent and high-quality words or word pairs from the large prompt domain generated by APE. The setup involves 10 agents, each sampling 5 prompts per round for 50 iterations. To ensure a fair comparison with 50 validation queries, we pair agents and take the maximum score among the prompts they generate as the final performance of FedOne.

**Preference Feedback.** For methods based on preference feedback, including FedPOB-Pref, FLDB-OGD, FLDB-GD, APOHF, and Double-TS, each iteration samples two prompts and queries the validation set twice to obtain a pairwise preference. Running for 50 iterations therefore requires 100 validation queries in total. We report the best reward at the 50th iteration (based on 100 queries in total). Other hyperparameters follow their original settings to ensure a fair comparison.

## D    MORE EXPERIMENTAL RESULTS

### D.1    ADDITIONAL EXPERIMENTS ON PROMPT DOMAIN GENERATION METHODS

**Performance and Stability Across Different Prompt Domains.** In the experiment section, we use GPT-3.5-Turbo to generate the prompt domain via APE. To further validate that our algorithm achieves superior performance across different prompt domains generated by different methods, we replace GPT-3.5-Turbo with GPT-4o-mini while keeping all other settings fixed, such as running both our algorithm and the baselines under the same LLM model, GPT-3.5-Turbo. As shown in Fig. 9, Our method consistently achieves strong performance across different prompt domains, underscoring its robustness to domain variability. Beyond maintaining high accuracy, it is capable of identifying near-optimal prompts in a sample-efficient manner, thereby reducing the overall cost of API queries to LLMs.

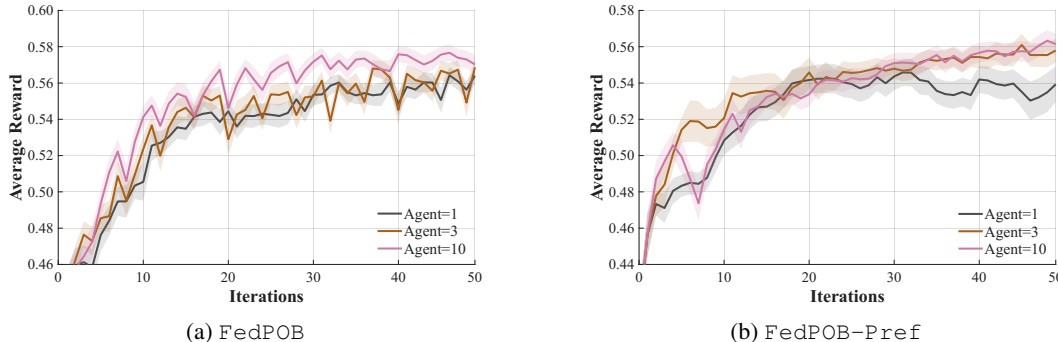

(a) FedPOB

(b) FedPOB-Pref

Figure 9: Performance across different prompt domains

## D.2 COMPLETE RESULTS IN FEDPOB-PREF

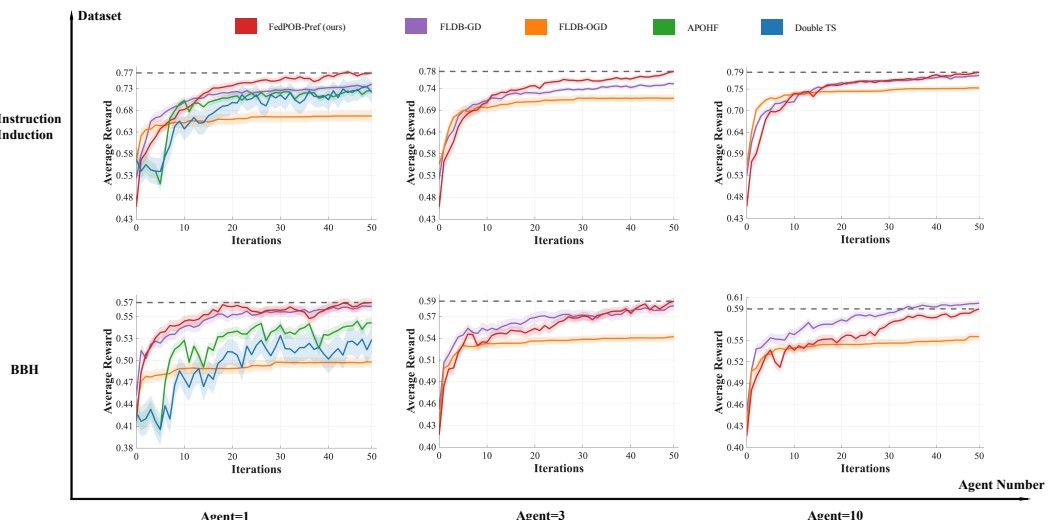

Figure 10: More detailed comparison for FedPOB-Pref using GPT-3.5-Turbo.

## D.3 COMPLETE RESULTS FOR FEDPOB

Table 5: Performance comparison on the complete set of Instruction Induction tasks.

| Dataset | INSTINCT | PromptBreeder | FedOne (10 agents) | FedPOB | | |
|---|---|---|---|---|---|---|
| | | | | 1 Agent | 3 Agents | 10 Agents |
| Active to Passive | 0.940±0.053 | **1.000±0.000** | **1.000±0.000** | 0.804±0.160 | 0.960±0.014 | 0.972±0.023 |
| Auto Categorization | 0.313±0.012 | 0.220±0.020 | 0.264±0.004 | 0.272±0.030 | **0.308±0.018** | 0.288±0.023 |
| Antonyms | 0.767±0.023 | 0.840±0.020 | **0.870±0.005** | 0.792±0.046 | 0.812±0.027 | 0.828±0.023 |
| Common Concept | 0.217±0.040 | 0.118±0.010 | 0.136±0.003 | 0.188±0.015 | **0.210±0.007** | 0.208±0.018 |
| Diff | **1.000±0.000** | **1.000±0.000** | **1.000±0.000** | 0.992±0.018 | **1.000±0.000** | **1.000±0.000** |
| First Word Letter | **1.000±0.000** | 1.000±1.000 | 0.713±0.089 | 1.000±1.000 | **1.000±1.000** | **1.000±1.000** |
| Informal to Formal | 0.570±0.020 | 0.521±0.067 | **0.605±0.005** | 0.528±0.028 | 0.528±0.039 | 0.570±0.030 |
| Larger Animal | **0.993±0.012** | 0.987±0.012 | 0.829±0.037 | 0.984±0.017 | 0.992±0.011 | 0.989±0.011 |
| Letters List | **1.000±0.000** | **1.000±0.000** | 0.831±0.095 | 0.952±0.107 | **1.000±0.000** | **1.000±0.000** |
| Negation | 0.860±0.020 | **0.927±0.012** | 0.897±0.010 | 0.856±0.061 | 0.940±0.014 | 0.920±0.032 |
| Num to Verbal | **1.000±0.000** | **1.000±0.000** | **1.000±0.000** | **1.000±0.000** | **1.000±0.000** | **1.000±0.000** |
| Orthography Starts With | 0.767±0.214 | 0.813±0.061 | 0.436±0.024 | 0.804±0.100 | 0.828±0.056 | **0.832±0.087** |
| Rhymes | 0.493±0.142 | 0.393±0.031 | **0.916±0.027** | 0.664±0.120 | 0.776±0.187 | 0.844±0.106 |
| Second Word Letter | 0.847±0.110 | 0.947±0.042 | 0.625±0.034 | 0.792±0.199 | 0.880±0.157 | **0.972±0.023** |
| Sentence Similarity | 0.467±0.031 | 0.380±0.020 | 0.360±0.035 | **0.540±0.094** | 0.508±0.082 | 0.448±0.018 |
| Sentiment | 0.973±0.012 | 0.993±0.012 | **0.996±0.002** | 0.988±0.018 | 0.972±0.023 | 0.972±0.027 |
| Singular to Plural | 0.993±0.012 | **1.000±0.000** | **1.000±0.000** | **1.000±0.000** | 0.996±0.009 | **1.000±0.000** |
| Sum | **1.000±0.000** | **1.000±0.000** | **1.000±0.000** | 0.984±0.036 | **1.000±0.000** | **1.000±0.000** |
| Synonyms | 0.327±0.150 | 0.333±0.115 | 0.320±0.023 | 0.324±0.103 | 0.296±0.041 | **0.384±0.124** |
| Taxonomy Animal | 0.947±0.023 | **0.967±0.042** | 0.805±0.026 | 0.924±0.073 | 0.980±0.024 | 0.972±0.034 |
| Translation En-De | 0.820±0.020 | 0.820±0.060 | **0.927±0.004** | 0.820±0.047 | 0.840±0.032 | 0.868±0.036 |
| Translation En-Es | 0.747±0.042 | 0.746±0.023 | **0.950±0.012** | 0.756±0.026 | 0.740±0.072 | 0.728±0.030 |
| Translation En-Fr | **0.947±0.023** | 0.920±0.040 | 0.919±0.005 | 0.944±0.033 | 0.940±0.283 | 0.948±0.018 |
| Word in Context | 0.553±0.058 | **0.620±0.040** | 0.409±0.091 | 0.460±0.084 | 0.640±0.020 | 0.608±0.036 |
| Object Counting | 0.520±0.106 | 0.473±0.110 | 0.497±0.019 | 0.520±0.074 | **0.616±0.039** | 0.588±0.050 |
| Odd One Out | **0.867±0.058** | 0.833±0.116 | 0.859±0.024 | 0.800±0.122 | 0.900±0.000 | **0.900±0.000** |
| Periodic Elements | **1.000±0.000** | **1.000±0.000** | 0.946±0.017 | 0.976±0.054 | **1.000±0.000** | **1.000±0.000** |
| Word Sorting | 0.753±0.058 | 0.753±0.099 | 0.497±0.026 | 0.756±0.093 | 0.744±0.065 | **0.828±0.063** |
| Word Unscrambling | 0.687±0.012 | 0.687±0.023 | **0.728±0.005** | 0.724±0.046 | 0.716±0.026 | 0.720±0.028 |
| Average 29 Task | 0.7715 | 0.7687 | 0.7356 | 0.7637 | 0.7977 | **0.8068** |

## D.4 FURTHER EVALUATION ACROSS LLM MODELS

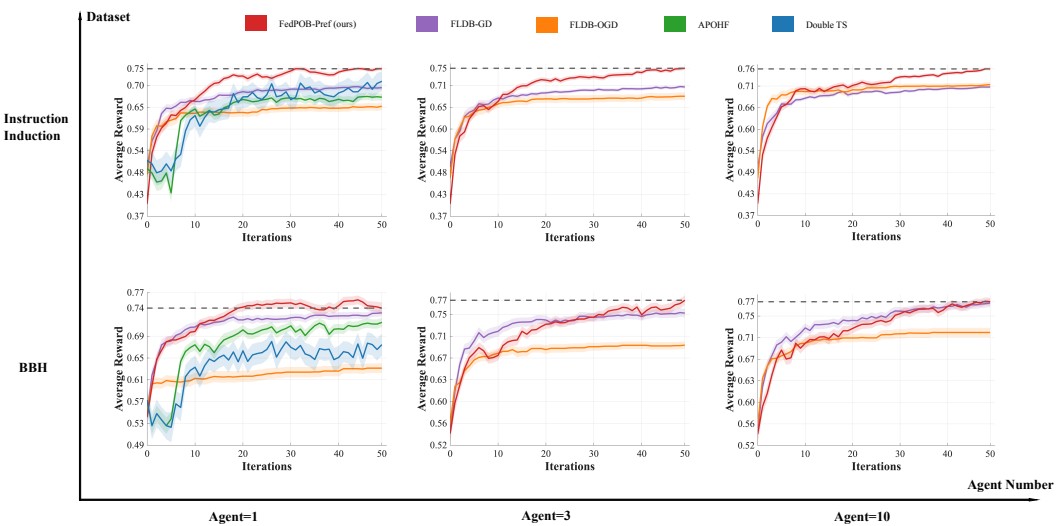

Figure 11: More detailed comparison for `FedPOB-Pref` using GPT-4o-mini.

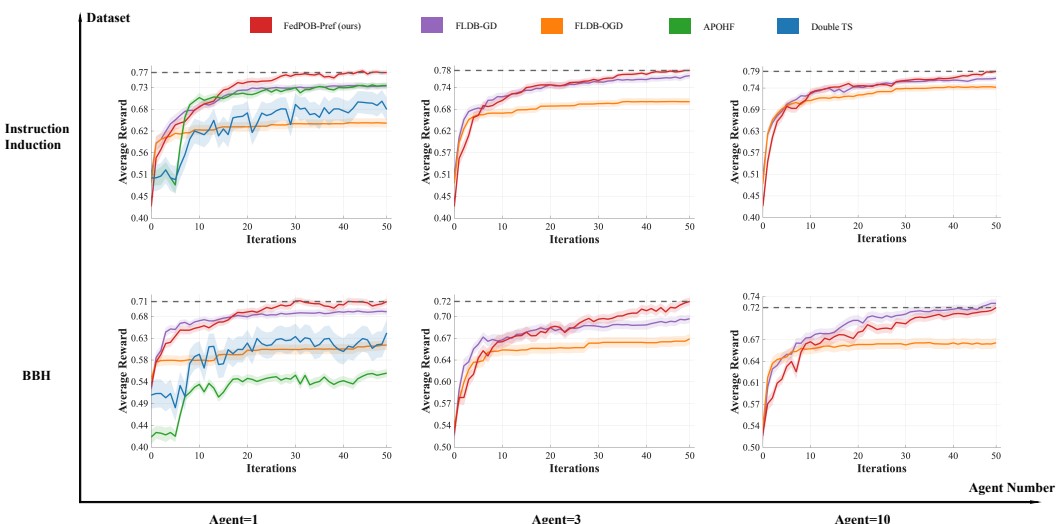

Figure 12: More detailed comparison for `FedPOB-Pref` using Qwen3-235B-A22B-2507.

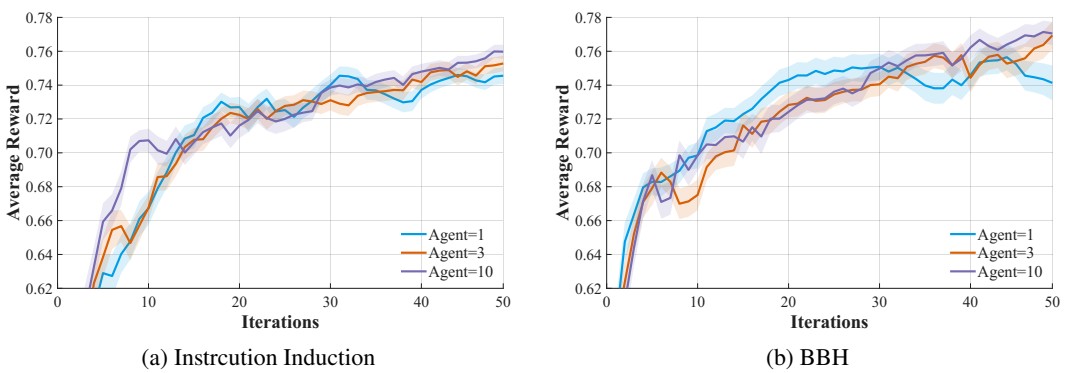

Figure 13: The performance of `FedPOB-Pref` across different iterations GPT-4o-mini.

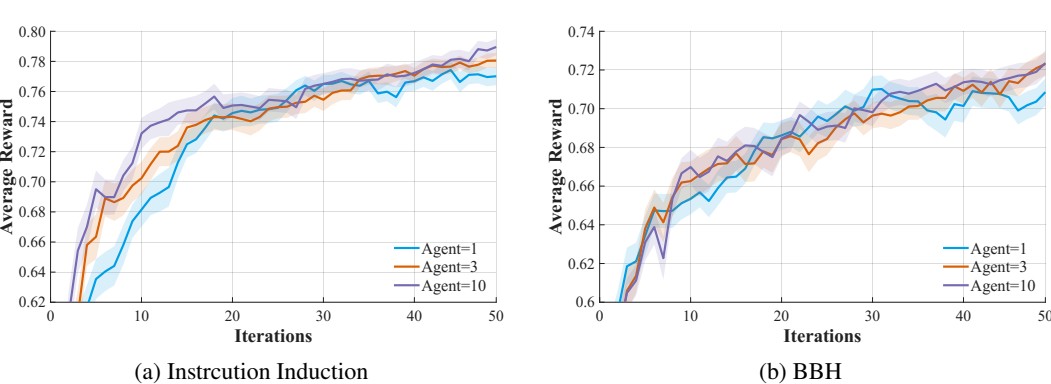

Figure 14: The performance of `FedPOB-Pref` across different iterations Qwen3-235B-A22B-2507.

# E  MATHEMATICAL PRINCIPLES OF THE LOCAL OBJECTIVE FUNCTION ADOPTED BY FEDPOB-PREF

This section provides a rigorous mathematical analysis of the local objective function adopted by FedPOB-Pref for federated optimization. We derive the first-order optimality conditions and demonstrate the necessity of the linear dual term for ensuring convergence to a globally optimal and consistent solution. The results here provide theoretical support for the design of our FedPOB-Pref algorithm.

## E.1  PROBLEM FORMULATION

The standard federated learning objective is to minimize a global function $F(\theta)$, defined as the average of $m$ local client objectives $f_i : \mathbb{R}^d \to \mathbb{R}$:

$$F(\theta) = \frac{1}{m} \sum_{i=1}^{m} f_i(\theta).$$

For distributed optimization, this is equivalently formulated as a constrained problem with local variables $\theta_i$ and a global consensus variable $\theta$:

$$\min_{\theta, \{\theta_i\}_{i=1}^m} \frac{1}{m} \sum_{i=1}^{m} f_i(\theta_i) \quad \text{s.t.} \quad \theta_i - \theta = 0, \quad \forall i \in \{1, \dots, m\}. \tag{6}$$

## E.2  THE AUGMENTED LAGRANGIAN METHOD

The constrained problem in Eq. equation 6 can be solved using the Method of Multipliers. We introduce a dual variable (Lagrange multiplier) $a_i \in \mathbb{R}^d$ for each consensus constraint and add a quadratic penalty term for the constraint violation. This forms the augmented Lagrangian function $\mathcal{L}$:

$$\mathcal{L}(\{\theta_i\}, \theta, \{a_i\}) = \frac{1}{m} \sum_{i=1}^{m} f_i(\theta_i) + \sum_{i=1}^{m} \langle a_i, \theta_i - \theta \rangle + \frac{\gamma}{2} \sum_{i=1}^{m} \|\theta_i - \theta\|^2,$$

where $\gamma > 0$ is a penalty parameter. An iterative algorithm then seeks a saddle point of this function.

## E.3  FIRST-ORDER STATIONARITY CONDITIONS

A stationary point of the augmented Lagrangian must satisfy $\nabla_{\theta_i} \mathcal{L} = 0$ and $\nabla_\theta \mathcal{L} = 0$. These first-order conditions are derived as follows.

The partial derivative with respect to a local variable $\theta_i$ is:

$$\frac{\partial \mathcal{L}}{\partial \theta_i} = \frac{1}{m} \nabla f_i(\theta_i) + a_i + \gamma(\theta_i - \theta) = 0. \tag{7}$$

The partial derivative with respect to the global variable $\theta$ is:

$$\frac{\partial \mathcal{L}}{\partial \theta} = -\sum_{i=1}^{m} a_i - \gamma \sum_{i=1}^{m} (\theta_i - \theta) = 0 \quad \Longrightarrow \quad \sum_{i=1}^{m} a_i = -\gamma \sum_{i=1}^{m} (\theta_i - \theta). \tag{8}$$

To see the implication of these conditions, we sum Eq. equation 7 over all clients $i$:

$$\frac{1}{m} \sum_{i=1}^{m} \nabla f_i(\theta_i) + \sum_{i=1}^{m} a_i + \gamma \sum_{i=1}^{m} (\theta_i - \theta) = 0.$$

Substituting the expression for $\sum_i a_i$ from Eq. equation 8 into the above yields:

$$\frac{1}{m} \sum_{i=1}^{m} \nabla f_i(\theta_i) - \gamma \sum_{i=1}^{m} (\theta_i - \theta) + \gamma \sum_{i=1}^{m} (\theta_i - \theta) = 0,$$

which simplifies to:

$$\frac{1}{m} \sum_{i=1}^{m} \nabla f_i(\theta_i) = 0.$$

This proves that any stationary point of $\mathcal{L}$ satisfies that the average of the local gradients is zero. If the solution is also primally feasible (i.e., $\theta_i = \theta$), this condition becomes precisely the first-order optimality condition for the original global problem:

$$\frac{1}{m} \sum_{i=1}^{m} \nabla f_i(\theta) = 0 \quad \Longleftrightarrow \quad \nabla F(\theta) = 0.$$

### E.4   ANALYSIS OF THE FORMULATION

#### E.4.1   PROOF OF NECESSITY FOR THE LINEAR DUAL TERM

To prove that the linear term $\langle a_i, \theta_i - \theta \rangle$ is necessary, we analyze the case where it is omitted, relying solely on a quadratic penalty. The objective would be:

$$\tilde{\mathcal{L}} = \frac{1}{m} \sum_i f_i(\theta_i) + \frac{\gamma}{2} \sum_i \|\theta_i - \theta\|^2.$$

The first-order condition with respect to $\theta_i$ for this objective is:

$$\frac{1}{m} \nabla f_i(\theta_i) + \gamma(\theta_i - \theta) = 0.$$

At a point of consensus where $\theta_i = \theta$ for all $i$, the penalty term vanishes, and the condition stringently requires that:

$$\frac{1}{m} \nabla f_i(\theta) = 0 \quad \Longrightarrow \quad \nabla f_i(\theta) = 0, \quad \forall i.$$

This is a significantly stronger condition than global optimality, as it requires the solution $\theta$ to be a stationary point for every client's objective function simultaneously. Such a point is generally non-existent for heterogeneous data distributions where local minima differ. Therefore, the inclusion of the linear dual term is mathematically essential to relax this condition to the correct global one, $\sum_i \nabla f_i(\theta) = 0$.

#### E.4.2   INTERPRETATION OF THE DUAL VARIABLES AT CONVERGENCE

In iterative methods that solve for a saddle point of $\mathcal{L}$, the dual variables are typically updated via dual ascent:

$$a_i^{t+1} = a_i^t + \gamma(\theta_i^{t+1} - \theta^{t+1}). \tag{9}$$

If the algorithm converges to a primally feasible solution $\theta^\star$, then $\lim_{t \to \infty}(\theta_i^{t+1} - \theta^{t+1}) = 0$. At this limit, the stationarity condition from Eq. equation 7 must hold. As $\theta_i \to \theta^\star$ and $\theta \to \theta^\star$, the equation implies that the dual variables converge to a fixed point $a_i^\star$:

$$\frac{1}{m} \nabla f_i(\theta^\star) + a_i^\star + \gamma(\theta^\star - \theta^\star) = 0 \quad \Longrightarrow \quad a_i^\star = -\frac{1}{m} \nabla f_i(\theta^\star).$$

This result provides a clear interpretation of the dual variable at the optimal solution: $a_i^\star$ is precisely the negative of the $i$-th client's scaled local gradient at the global optimum. The condition $\sum_i a_i^\star = 0$ (from Eq. equation 8 at convergence) then mathematically guarantees that $\sum_i \nabla f_i(\theta^\star) = 0$. The dual variables are thus the mechanism that allows local gradients to be non-zero while ensuring their sum is zero.

## F   OPTIMIZED PROMPTS FROM FEDPOB AND FEDPOB-PREF

In this section, we present the optimized prompts together with their validation-set scores obtained by our `FedPOB` and `FedPOB-Pref` across all 53 tasks in both the Instruction Induction and BBH datasets after 50 optimization rounds. For each task in the tables, the *upper row* reports the

prompt and score optimized by `FedPOB`, while the *lower row* corresponds to those optimized by `FedPOB-Pref`.

Table 6: Optimized prompts and their scores for the Instruction Induction tasks

| Task | Prompt | Score |
|------|--------|-------|
| active to Passive | Rewrite the sentence passively. | 0.972 |
| | The sentence should be changed to passive voice: "The sentence is to be changed from active to passive voice." | 0.993 |
| antonyms | change the prefix of the word to make it have the opposite meaning. | 0.288 |
| | find the opposite of each given word. | 0.293 |
| auto categorization | provide an appropriate category for each group of items. | 0.828 |
| | identify the category or group that each set of inputs belong to. | 0.840 |
| common concept | provide a connection between two seemingly unrelated words or phrases. | 0.208 |
| | provide a connection between two seemingly unrelated items. | 0.250 |
| diff | change the prefix of the word to make it have the opposite meaning. | 1.000 |
| | Find the disparity between the initial number and the subsequent number in every input. | 1.000 |
| first word letter | Return the initial letter of every word provided as input. | 1.000 |
| | State the initial letter of the specified word. | 1.000 |
| informal to formal | rephrase the given sentences, not just provide synonyms. Here are the revised sentences: Input: Can you complete all of these tasks? Output: Are you capable of completing all of these tasks? Input: It is not advisable to take any action at this time. Output: It is not recommended to do anything right now. Input: I'll see you this evening. Output: I anticipate seeing you tonight. Input: Would you like me to accompany you? Output: Do you want me to go along with you? Input: The entire narrative was fabricated. Output: The entire story was created. | 0.570 |
| | rephrase the sentences using different words or phrases with the same meaning. | 0.607 |
| larger animal | choose the animal with the larger size or more strength. | 0.989 |
| | choose the larger animal in each pair. | 1.000 |
| letters list | Add a space between each letter within a word. | 1.000 |
| | Show each individual letter of the given word with a space between each letter. | 1.000 |
| negation | change the sentences to negative form, indicating that the statements are false. | 0.920 |
| | change the statements to the opposite meaning. | 0.947 |
| num to verbal | Create a program that translates a provided number into its equivalent word form. | 1.000 |
| | Write out the number in words from one to nine thousand, nine hundred and ninety-nine. | 1.000 |
| object counting | count the total number of animals/items mentioned in the input sentence. | 0.588 |
| | count the number of items listed in the input. | 0.660 |
| odd one out | Find the word that is not the same as the others in the group. | 0.900 |
| | Select the word that is not related to the rest. | 1.000 |
| orthography start with | identify and output the word that starts with the specified letter. | 0.832 |
| | identify the word in the sentence that starts with the given letter. | 0.907 |
| periodic element | Give the names of the elements that match the provided atomic numbers. | 1.000 |
| | List the names of the elements corresponding to the provided atomic numbers. | 1.000 |
| rhymes | find a word that rhymes with the given word, so in the case of "buy", the output would be "buy" as it already rhymes with itself. | 0.844 |
| | change the first letter of the word to make a new word. | 0.993 |
| second word letter | Retrieve the second letter from the given word. | 0.972 |
| | Print the second-to-last letter of the input word. | 0.980 |
| sentence similarity | determine the likelihood that the two sentences are talking about the same topic. The outputs provided are the level of certainty in the similarity of the topics discussed in the sentences. | 0.448 |
| | compare the similarity between two sentences using a scale from 0 to 5, with 0 being "definitely not " similar and 5 being "perfectly " similar. The output provided for each pair of sentences indicates the level of similarity between them based on the comparison. | 0.613 |
| sentiment | classify the input as either positive or negative based on the given statement. | 0.972 |
| | provide an output (positive or negative) based on the given input. | 1.000 |
| singular to plural | pluralize the given input words. | 1.000 |
| | add the letter "s" to the end of the word. | 1.000 |
| sum | Calculate the total by adding the two numbers given as input. | 1.000 |
| | sum the two inputted numbers. | 1.000 |
| synonyms | provide alternative words for the given inputs. | 0.384 |

Table 6: Optimized prompts and their scores for the Instruction Induction tasks

| Task | Prompt | Score |
|------|--------|-------|
| | provide an antonym, synonym, or rhyme for the given word. | 0.500 |
| taxonomy animal | list the animals from the input words. | 0.972 |
| | List the animals from the given words. | 1.000 |
| translation en-de | Translate the specified words from English into German. | 0.868 |
| | Übersetze die gegebenen englischen Wörter ins Deutsche. | 0.887 |
| translation en-es | traduce cada palabra al español. | 0.728 |
| | Convert the following words from English to Spanish: 1. wardrobe - armario 2. care - preocuparse 3. dissatisfaction - insatisfacción 4. pond - estanque 5. trial - prueba | 0.807 |
| translation en-fr | translate the words provided from the English language to French. | 0.948 |
| | turn the words into French. | 0.960 |
| word in context | determine if the word is used in the same context in both sentences. In this case, the word "academy" is used in different contexts in the two sentences, so the output is "not the same." | 0.608 |
| | determine if the two sentences provided have the same meaning based on the given word. | 0.700 |
| word sorting | sort the words in the provided list in alphabetical order. Each output should be a single line of the sorted words, separated by spaces. | 0.828 |
| | rearrange the words in the list in alphabetical order. | 0.867 |
| word unscrambling | Solve the jumbled words provided. | 0.720 |
| | Arrange the scrambled words in the correct order. | 0.793 |

Table 7: Optimized prompts and their scores for the BBH tasks

| Task | Prompt | Score |
|------|--------|-------|
| boolean expressions | Assess the provided logical expressions and produce the result. | 0.844 |
| | Assess the provided logical expressions and give the resulting output. | 0.860 |
| date understanding | determine the date a specific number of days or years ago from a given date. | 0.572 |
| | determine the date one week ago or one week from today based on the given information. | 0.613 |
| disambiguation qa | identify the antecedent of the pronoun in each sentence or state if it is ambiguous. The correct antecedent for each sentence is as follows: '1. (C) Ambiguous 2. (B) The office was Sam's office 3. (A) The technician completed the repair 4. (A) Alex could not meet 5. (B) Asked the cleaner | 0.840 |
| | explain the antecedent of the pronoun in the given sentences or state if it is ambiguous. The correct antecedent for each sentence is provided in the output. | 0.793 |
| dyck languages | Finish the remaining part of the series and ensure that all parentheses are closed correctly. | 0.680 |
| | Continue the sequence, ensuring that all parentheses are closed correctly. | 0.740 |
| formal fallacies | determine if the argument, given the explicitly stated premises, is deductively valid or invalid. The output for all the provided inputs is "invalid." | 0.812 |
| | determine whether the arguments, given the explicitly stated premises, are deductively valid or invalid. | 1.000 |
| geometric shapes | Identify the geometric shape represented by the given SVG path element, with the provided outputs indicating the corresponding shape based on the paths. | 0.448 |
| | Determine the shape illustrated by the given SVG path element. | 0.487 |
| hyperbaton | choose the sentence with the correct adjective order, which is the order of opinion, size, age, shape, color, origin, material, and purpose. | 0.948 |
| | choose the sentence with the correct adjective order. | 0.973 |
| logical deduction five objects | determine which object is in a specific position in the given set of objects based on the information provided in each paragraph. | 0.476 |
| | determine which object finished first in each scenario. The correct outputs are: 1. (C) Ada finished first 2. (E) The falcon is the third from the left 3. (E) Amy finished first 4. (D) The plums are the second-cheapest 5. (D) The orange book is the third from the left. | 0.473 |
| logical deduction seven objects | determine which object is in a specific position in the set of seven objects based on the given statements. | 0.488 |
| | determine which object is in a specific position in the given arrangement of objects. | 0.540 |
| logical deduction three objects | determine which object is in a specific position based on the given information. In each case, the correct output is provided based on the logical consistency of the statements within the paragraph. | 0.644 |
| | determine which object is in the leftmost position based on the given information. | 0.653 |

*Continued on next page*

Table 7: Optimized prompts and their scores for the BBH tasks

| Task | Prompt | Score |
|---|---|---|
| movie recommendation | find a movie similar to the given list of movies. The correct options are selected based on the similarity to the movies listed in the input. | 0.732 |
| | find a movie similar to a given list of movies. The correct option for each set of movies is as follows: 1. (C) The Usual Suspects 2. (D) Fargo 3. (A) Pulp Fiction 4. (B) The Matrix 5. (A) Schindler's List | 0.780 |
| multistep arithmetic two | Find the difference between the first set of parentheses and the second set, and then simplify the expression. | 0.692 |
| | determine the outcome of the provided mathematical equation. | 0.700 |
| navigate | "Turn right. Take 10 steps. Turn around. Take 10 steps." | 0.716 |
| | take 9 steps left, then 10 steps forward, then 9 steps right, and finally 10 steps backward. By following these instructions, you would return to the starting point, so the output is Yes. | 0.773 |
| penguins in a table | determine specific information based on the given table of penguins and provide the correct answer from the options provided. | 0.605 |
| | determine specific information about the penguins based on the given data and answer the questions accordingly. | 0.586 |
| reasoning about colored objects | determine the color or quantity of items based on their arrangement in a row. | 0.568 |
| | determine the color of the item directly to the right of a specified color in a given arrangement of items. | 0.580 |
| ruin names | identify the humorous edit of the artist or movie name, and the correct answer for each input is provided in the output. | 0.724 |
| | find the humorous edit of the artist or movie name. | 0.813 |
| salient translation error detection | Find the mistake in the given translations. | 0.600 |
| | Find the mistake in the German to English translations given. | 0.613 |
| snarks | identify the sarcastic statement from the given options. The selected statement typically conveys an opposite meaning or is exaggerated in a way that highlights the absurdity of the situation. | 0.782 |
| | identify the sarcastic statement from the given options. In each case, the sarcastic statement is one that implies the opposite of what it literally says, often highlighting absurdity or exaggeration. | 0.793 |
| sports understanding | determine if the sentences were plausible based on common sports terminology. | 0.564 |
| | determine if the sentences provided are plausible in a sports context. | 0.580 |
| temporal sequence | determine between what times the person could have gone to the specified location based on the given information about their activities throughout the day. The correct time range is then provided as the output. | 0.652 |
| | determine between what times the person could have gone to a specific location based on the given information. The correct options for each scenario are as follows: 1. David could have gone to the construction site between 8am to 12pm (Option A). 2. Leslie could have gone to the market between 11am to 5pm (Option B). | 0.700 |
| tracking shuffled objects five objects | determine who Claire is dancing with at the end of the dance. In the given scenario, at the end of the dance, Claire is dancing with option (B) Sam. | 0.328 |
| | determine who ends up with a specific item or partner after a series of swaps or trades. | 0.353 |
| tracking shuffled objects seven objects | determine the final position/book/ball of a specific person/player after a series of swaps. | 0.256 |
| | determine the final partner, gift, ball, or book that a specific person has at the end of the given scenario. | 0.293 |
| tracking shuffled objects three objects | determine the final position or item that Bob ends up with after a series of swaps. | 0.400 |
| | determine who ends up with a specific item after a series of swaps in a white elephant gift exchange. | 0.433 |
| web of lies | determine if Inga tells the truth based on the statements given by the other individuals. In this case, the answer is "No" because Inga says Fidel tells the truth, but Fidel says Vernell lies. Since there is a contradiction in the statements, Inga does not tell the truth. | 0.636 |
| | determine if Christie tells the truth based on the statements of the other individuals. Christie says that Teressa tells the truth. Since Teressa says that Leda lies, and Leda says that Shaunda lies, and Shaunda says that Ryan tells the truth, we can conclude that Christie is telling the truth. | 0.667 |

# G ADDITIONAL ABLATION STUDY

## G.1 EVALUATION UNDER HETEROGENEOUS AGENT CAPABILITIES

Since real-world deployments often involve agents equipped with LLMs of widely varying capacities, it is important to assess whether weaker models can destabilize the optimization process. To evaluate FedPOB's robustness under such heterogeneous-agent settings, we constructed a mixed-model setup using three LLMs with different capability levels. These models were combined into two representative mixed-model groups:

- **Group 1**: GPT-3.5-turbo + Qwen3-235B + GPT-5-nano
- **Group 2**: Llama-3.2-1B + Qwen3-235B + GPT-5-nano

For the 3-agent setting, we directly mix the three models. For the 10-agent setting, we include four weaker agents (GPT-3.5-turbo or Llama-3.2-1B) and three stronger agents.

Since different LLMs exhibit different score distributions, instead of averaging rewards across all agents, we report the best reward of the mid-strength agent (Qwen3-235B, Fig. 15a) and the strongest agent (GPT-5-nano, Fig. 15b) separately at iteration 50.

- **Group 1: Performance steadily improves as the number of agents increases.**
  GPT-3.5-turbo—although weaker—still contributes meaningful exploration signals, demonstrating that FedPOB can effectively handle heterogeneous agents.(See Fig. 15c)
- **Group 2: Performance does not increase monitonously when more agents are added.**
  We find that llama-3.2-1B performs extremely poorly on the prompt-optimization task. When multiple such weak agents participate, their noisy local updates can negatively affect the stronger models.(See Fig. 15d)

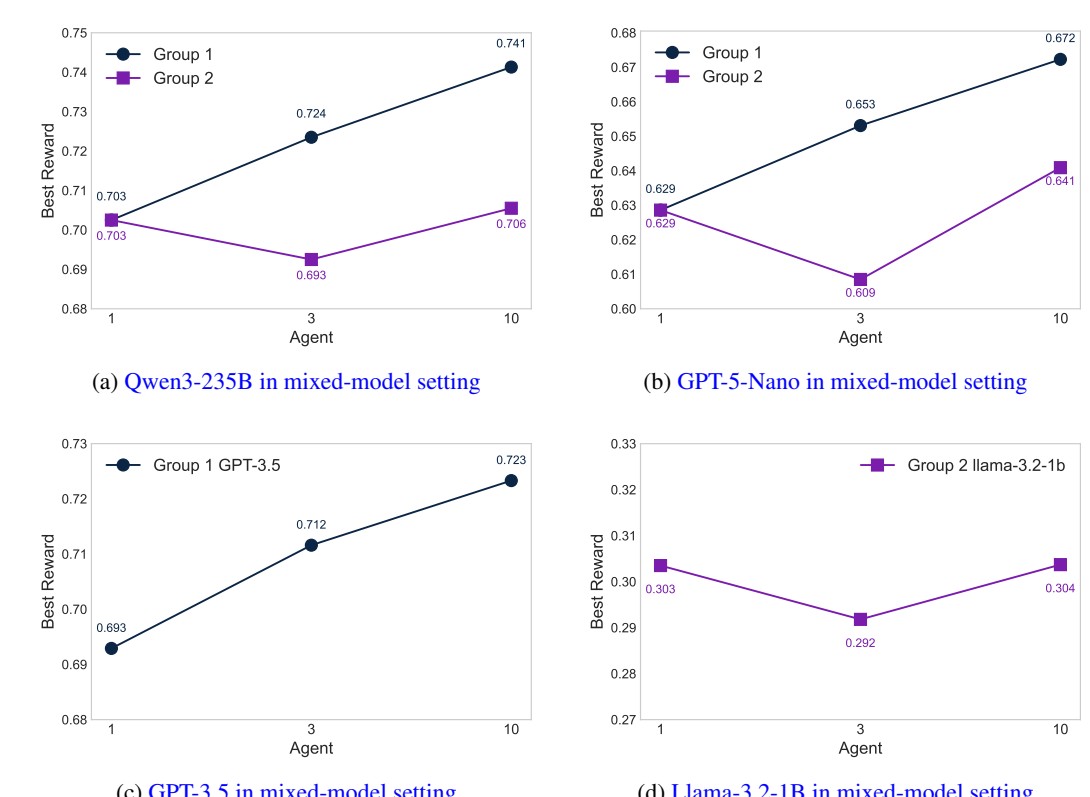

(a) Qwen3-235B in mixed-model setting

(b) GPT-5-Nano in mixed-model setting

(c) GPT-3.5 in mixed-model setting

(d) Llama-3.2-1B in mixed-model setting

Figure 15: Performance of heterogeneous mixed-model configurations across four representative LLMs.

Overall, these results show that FedPOB does not require agents to have similar capabilities, even moderately weaker models can contribute useful exploration and improve performance. But extremely weak models may hinder performance when they participate in the optimization process.

## G.2 TESTING THE REASONABLENESS OF LINEAR MODELS

### G.2.1 REASONABLENESS OF LINEAR MODELS

To empirically justify the use of a linear model, we conducted a study to assess whether the mapping from prompt embeddings to performance scores can be well-approximated by a linear function, or whether a non-linear model provides substantial additional benefit.

**The Experimental Setup is shown below:**

- **Objective:** To compare the predictive performance of a linear regressor versus a non-linear neural network regressor.
- **Data:** For each of the 53 tasks in our study, we used the full set of 500 prompt embeddings as the input features ($X$) and their corresponding validation scores as the target variable ($y$).
- **Linear Model:** We used L2-regularized linear regression (Ridge Regression). This model directly corresponds to the linear model used by our algorithms.
- **Non-linear Model:** We used a Multi-Layer Perceptron (MLP) regressor with L2 regularization. To ensure fair comparison, we adopted the same MLP architecture as APOHF (Lin et al., 2024a) which used the same embedding model as our work.

**Evaluation Protocol.** To ensure a fair and robust comparison in the high-dimensional setting, we evaluated both the L2-regularized linear model and the L2-regularized neural network using K-fold cross-validation (CV) with $K = 5$. The L2 regularization hyperparameter for each model was tuned via a nested CV loop exclusively on the training data within each outer fold. We report two metrics here:

1. **Cross-Validated $R^2$ ($R^2_{\text{CV}}$):** The primary metric for generalization and predictive power on unseen data, calculated from the out-of-sample predictions on the held-out validation folds.
2. **In-Sample (Training) $R^2$:** The $R^2$ score averaged across the training folds, used to assess the degree of overfitting.

We averaged the $R^2$ scores across all 53 tasks, and the results are shown in the table below.

Table 8: Comparison of Linear and Non-Linear Models in Predicting Prompt Performance

| Model | $R^2_{\text{CV}}$ (Predictive Power) | $R^2$ (In-sample Fit) |
|---|---|---|
| **Linear Model** | 0.502 | 0.740 |
| **Neural Network (MLP)** | 0.452 | 0.799 |

The results show that while the neural network achieves a higher *in-sample $R^2$* (0.799 vs. 0.740), its *cross-validated $R^2$* is notably lower than the linear model's (0.452 vs. 0.502). This indicates that the non-linear neural network model is overfitting the training data and generalizes more poorly than the simple linear model. These results strongly suggest that a linear model is not significantly mis-specified and is a suitable choice for this problem. In addition, the reasonableness of linear models is further justified by the outstanding performance of our methods in more than 50 tasks.

### G.2.2 OUR METHODS ARE ROBUST AGAINST EMBEDDING MIS-SPECIFICATION

To evaluate the robustness of our method under embedding mis-specification, we conducted an ablation study where we deliberately distorted the prompt embeddings. Specifically, we truncated each

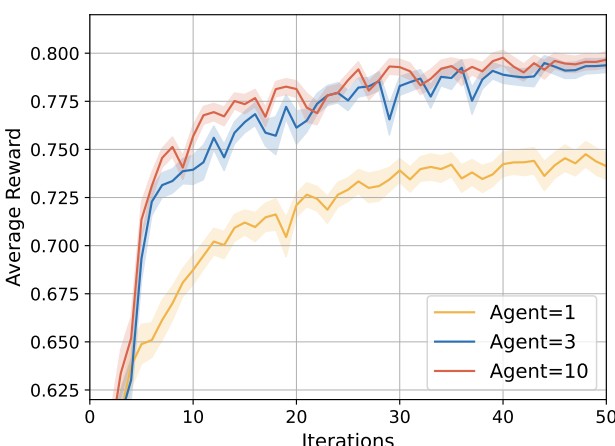

Figure 16: FedPOB performance with embedding misspecification.

768-dimensional embedding vector by removing its last $768/2 = 384$ dimensions and then renormalized the resulting vectors. This produces a mis-specified scenario as the reviewer has suggested.

The results (Fig. 16) show that:

1. **For 1-agent and 3-agent settings**, the performance remains nearly unchanged compared to the original results in Fig. 2.

2. **For the 10-agent setting**, the distorted embedding leads to a small performance drop, indicating that FedPOB becomes slightly more sensitive to model mis-specification when more agents are involved.

3. **Across all settings**, the overall trend is preserved: increasing the number of agents generally improves performance.

These findings suggest that FedPOB maintains relatively stable behavior even under substantial embedding mis-specification, and the benefits of more participating agents continue to hold.

### G.2.3 COMPARISON WITH NEURAL BANDIT BASELINES

To compare our linear bandit–based approach with non-linear alternatives for federated black-box prompt optimization, we evaluated two families of neural bandit baselines across 29 Instruction-Induction tasks: federated neural bandits (score feedback) and federated neural dueling bandits (preference feedback).

1. **Score Feedback Baseline**: We compare against Federated Neural Bandit (Dai et al., 2023), a Neural-UCB–based federated bandit method. Since the original work does not include prompt optimization and its network architecture is incompatible with high-dimensional features (768-d embeddings), we adapt the neural architecture from APOHF (Lin et al., 2024a), which uses the same embedding model as ours and ensures a fair, architecture-aligned comparison.

2. **Preference Feedback Baseline**: For preference feedback, we extend APOHF, a neural dueling bandit method for prompt optimization, to the federated setting. To ensure consistency and fairness, we adopt the FedAvg aggregation protocol, following Federated Neural Bandit.

3. **Results**: We report the best rewards across all 29 Instruction-Induction tasks under both score-feedback and preference-feedback settings, comparing FedPOB and FedPOB-Pref with their corresponding neural baselines in Fig. 17.

Across both feedback types, compared with our FedPOB and FedPOB-Pref, the federated neural bandit and neural dueling bandit baselines fail to scale effectively. Their performance does not

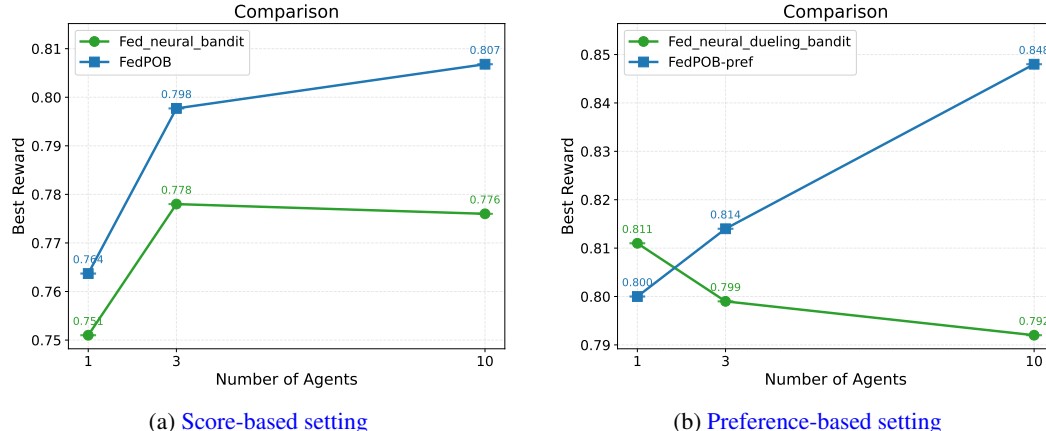

(a) Score-based setting            (b) Preference-based setting

Figure 17: Comparison with neural bandit baselines under score-based and preference-based feedback.

consistently improve as the number of agents increases. This may be due to the non-convexity of neural networks combined with data heterogeneity: local neural updates diverge across agents, making it difficult for standard federated averaging to converge to a stable global model in a small number of iterations. This in fact also aligns with theoretical results, since federated neural bandit (Dai et al., 2023) is not guaranteed to achieve better performance with more agents. In contrast, the performance of federated linear bandits is theoretically guaranteed to improve as the number of agents increases, which is one of our major motivations for adopting linear models.

**Conclusion:** These new findings together justify that the linear model is a reasonable and robust choice in our problem, and performs better than non-linear bandits.

### G.3 PARAMETERS EXCHANGED BETWEEN AGENTS AND THE SERVER

To better demonstrate our communication efficiency, we present additional results below to show that the total number of triggers (i.e., communication rounds) and the total throughput are small for both FedPOB and FedPOB-Pref. For throughput, we simply measure the total number of parameters exchanged between agents and the server over 50 iterations.

#### G.3.1 SCORE FEEDBACK SETTING (FEDPOB)

In FedPOB, we set communication threshold $D$ to limit unnecessary communication rounds unless local agents have collected enough information. The table below reports the total number of communication rounds for different values of $D$ over 50 iterations of FedPOB (average across all Instruction-Induction tasks). To further quantify throughput, we compute the total number of parameters exchanged between the agents and the server over 50 iterations.

Table 9: Communication rounds for different values of D

| $D$ | 0 | 10 | 100 | 300 | 1000 |
|---|---|---|---|---|---|
| **# Comm. Rounds (Agent=3)** | 50 | 12 | 4 | 2 | 1 |
| **# Comm. Rounds (Agent=10)** | 50 | 11.13 | 3.99 | 2 | 1 |
| **Throughput (Agent=3)** | 169.0M | 40.56M | 13.52M | 6.75M | 3.37M |
| **Throughput (Agent=10)** | 563.2M | 125.4M | 44.94M | 22.53M | 11.26M |

As expected in Table 9, the frequency of communication declines sharply as the communication threshold $D$ increases. Fig. 4 demonstrates that even when $D$ is raised to 300 or 1000—limiting communication to merely two rounds or a single round, respectively—our method remains highly robust. It exhibits only a negligible reduction in reward, despite the substantial decrease in communication overhead.

### G.3.2 Preference Feedback Setting FedPOB-Pref

In the preference-feedback setting, we similarly evaluate communication efficiency by examining both the number of rounds and total throughput. We compare FedPOB-Pref to FLDB-GD and FLDB-OGD (Table 10 and Table 11) and observe:

- FedPOB-Pref outperforms FLDB-OGD by 5.9%, while using the same number of communication rounds.

- FedPOB-Pref matches FLDB-GD's performance while requiring only 1/30 of its communication rounds.

- FedPOB-Pref remains robust even under much stricter communication constraints. When we allow only one communication trigger every three iterations, the total number of communication rounds and the overall throughput are reduced by 68%. Despite this substantial reduction, the performance decreases by only 4.06% and 7.07% for 3 and 10 agents, respectively. This demonstrates that FedPOB-Pref provides a reliable option in scenarios where communication resources are highly limited or expensive.

Table 10: Performance comparison of different methods (Agent = 3).

| Method | Best Rewards | Comm. Times | Throughput |
|---|---|---|---|
| FedPOB-Pref | 0.8145 | 50 | 169.1M |
| FedPOB-Pref (Less Comm.) | 0.7814 | 16 | 54.11M |
| FLDB-GD | 0.7959 | 1500 | 175.3M |
| FLDB-OGD | 0.7687 | 50 | 169.0M |

Table 11: Performance comparison of different methods (Agent = 10).

| Method | Best Rewards | Comm. Times | Throughput |
|---|---|---|---|
| FedPOB-Pref | 0.8482 | 50 | 563.9M |
| FedPOB-Pref (Less Comm.) | 0.7882 | 16 | 180.4M |
| FLDB-GD | 0.8244 | 1500 | 584.4M |
| FLDB-OGD | 0.8123 | 50 | 563.5M |

### G.4 Scaling Law with Agents

To demonstrate the scalability of our methods to larger numbers of agents and reveal potential scaling laws, we adopted the Instruction-Induction tasks (29 tasks), and increased the number of agents to 25 and 100 under the same experimental setting (using gpt-3.5-turbo). The results in Fig. 18 summarize the final aggregated scores over all 29 tasks after 50 iterations, as well as the average reward trajectories across iterations.

These results suggest a clear scaling pattern:

- Our method is indeed able to scale to a substantially larger number of agents (100).

- The performance of FedPOB improves consistently and monotonically as the number of agents increases.

- As the number of agents increases, the marginal gain gradually diminishes. This suggests a logarithmic-like scaling curve where performance gains eventually saturate, which is a desirable and expected property in large-scale collaborative learning systems.

### G.5 Results with Heterogeneity in Tasks

To simulate a realistic scenario in which agents share a common prompt pool but operate on heterogeneous tasks, we conduct an additional experiment where all agents use the same prompt set

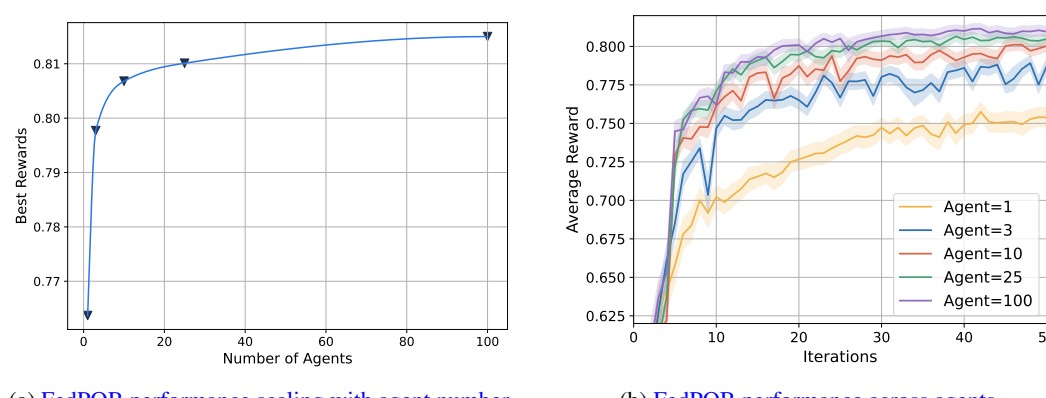

(a) FedPOB performance scaling with agent number.  (b) FedPOB performance across agents.

Figure 18: Scalability of FedPOB: performance improves consistently with more agents.

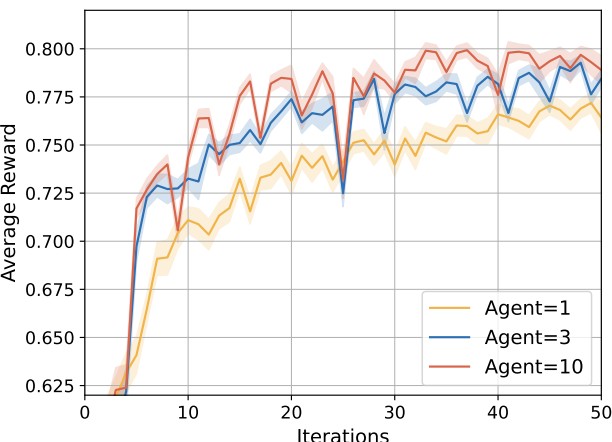

Figure 19: FedPOB performance under heterogeneous task settings

(containing 500 prompts), while their respective tasks (i.e., reward functions) remain distinct. Concretely, we induce task heterogeneity by perturbing the underlying reward function for each agent: independent Gaussian noise is added to the original reward values, yielding a set of agent-specific reward functions.

We evaluated this setting across all 29 Instruction-Induction tasks using GPT-3.5-turbo (OpenAI, 2023a) . The Fig. 19 demonstrate that FedPOB continues to perform robustly under this heterogeneous-task scenario. Even when the agents are optimizing against heterogeneous reward functions, our algorithm successfully leverages collaboration to improve overall performance. This confirms that FedPOB is adaptable to different types of federated heterogeneity, whether in the prompt space or the task definition.

