# OpenReview forum: "FedPOB: Sample-Efficient Federated Prompt Optimization via Bandits"
_ICLR.cc/2026/Conference — ICLR 2026 Conference Desk Rejected Submission_

### Official Review · Reviewer_uKNR · 2025-10-29

**Soundness:** 3
**Presentation:** 3
**Contribution:** 3
**Rating:** 8
**Confidence:** 3

**Summary:**

This paper addresses the challenges of large language model (LLM) prompt optimization, particularly the challenges imposed by black-box access, the requirement for sample efficiency, and the necessity of privacy-preserving collaboration. To resolve these issues simultaneously, the authors introduce Federated Prompt Optimization via Bandits (FedPOB) which formalises the problem as a Multi-Arm Bandit problem and apply federated algorithms. The problem settings spans both scalar and comparative preference feedbac. Their experiments confirm that both the FedPOB algorithm and its comparative extension, FedPOB-Pref, significantly surpass existing baselines, with performance reliably increasing as more agents participate in the federated collaboration.

**Strengths:**

1. The problem setting the authors consider is well-formulated and clear. Accordingly the paper is well-written and easy-to-follow.
2. The FedPOB algorithm has a solid theoretical foundation from the standard Multi-Armed Bandits literature that offers convergence guarantees and shown to improve with more collaborating agents.
3. The authors extend the problem to the setting with preference feedback that is common for subjective tasks with LLMs such as creative writing.
4. Evaluations span many model families and standard benchmark tasks to provide reliable results.

**Weaknesses:**

1. The paper relies heavily on the linearity assumption which is not tested. Empirical evidence that support this assumption could further solidify the results of this work.
2. There are no theoretical guarantees for the preference feedback setting in Section 4.2. While the authors motivate FedPOB-Pref with inspirations from duelling bandits and federated learning, only experimental evidence is provided for its performance.
3. The preference feedback setting is constrained to the BTL model which has known limitations. Formulation where the pairwise comparison $P(p_{t,1} \succ p_{t,2})$ is modelled directly would make the approach more applicable and general.

**Questions:**

1. Could the authors provide concrete examples to the motivation of federated learning for prompt optimisation? While the motivation for prompt optimisation is clear and why the bandit formulation can resolve the challenges of LLMs being black-boxes and costly to query, I struggle to find a practical example for federated prompt optimisation.
2. Does FedPOB work asynchronously? Is there any point when an agent has to wait for a weight update from the Central Server? For example, let's assume that there are two agents. Agent 1 accumulates information quickly and send updates to the central server every round while Agent 2 accumulates information slowly and only reaches the threshold in Line 8 Algo 1 every 10 rounds. Can multiple updates happen from Agent 1 without any from Agent 2? How would such an extreme setting affect performance?
3. In line 243 the authors claim that in many practical applications only preference feedback is available or it is more applicable. What are exactly these applications?
4. How does the acquisition logic (Line 3-4 in Algo 3) relate to previous works on Duelling Bandits? For example, it is similar to BAI version of the algorithm presented in [1]. Would the algorithm work with other approaches from the literature?
5. In Section 4, does the number of samples observed and used for learning increases with the number of agents? Is the batch size fixed per agent or per iteration? For a consistent evaluation, it would be valuable to show performance as a function of number of samples seen by all the agents.

[1]: Sample-Efficient Alignment for LLMs, Liu et al. (2024)

---

> ### Author Response · Authors · 2025-11-21
> **Response to Reviewer uKNR [Part 1 / 4]**
>
> We sincerely thank the reviewer for their rigorous and positive evaluation and for offering feedback that has undoubtedly improved the overall quality of the paper.
>
> ---
>
> ### **Question**:
>
> > **[W1].** The paper relies heavily on the linearity assumption which is not tested. Empirical evidence that support this assumption could further solidify the results of this work.
>
> ### **Response**:
>
> To empirically justify the reasonableness of adopting linear models, we conducted a new study to determine **if the mapping from prompt embeddings to their performance scores is reasonably approximated by a linear model**, or if a non-linear model offers a significant advantage.
>
> **Experimental Setup:**
> - **Objective:** To compare the predictive performance of a linear regressor versus a non-linear neural network regressor.
> - **Data:** For each of the 53 tasks in our study, we used the full set of 500 prompt embeddings as the input features ($X$) and their corresponding validation scores as the target variable ($y$).
> - **Linear Model:** We used L2-regularized linear regression (Ridge Regression). This model directly corresponds to the linear model used by our algorithms.
> - **Non-linear Model:** We used a Multi-Layer Perceptron (MLP) regressor with L2 regularization. To ensure fair comparison, we adopted the same MLP architecture as APOHF (Lin et al., 2024a) which used the same embedding model as our work.
>
> **Evaluation Protocol:**
> To ensure a fair and robust comparison in the high-dimensional setting, we evaluated both the L2-regularized linear model and the L2-regularized neural network using K-fold cross-validation (CV) with $K=5$. The L2 regularization hyperparameter for each model was tuned via a nested CV loop exclusively on the training data within each outer fold. We report two metrics here:
> 1.  **Cross-Validated $R^2$ ($R^2_{\mathrm{CV}}$)**: The primary metric for generalization and predictive power on unseen data, calculated from the out-of-sample predictions on the held-out validation folds.
> 2.  **In-Sample (Training) $R^2$**: The $R^2$ score averaged across the training folds, used to assess the degree of overfitting.
>
> **Results:**
> We averaged the $R^2$ scores across all 53 tasks, shown in the table below.
>
> | Model | $R^2_{\text{CV}}$ (Predictive Power) | $R^2$ (In-sample Fit) |
> | :--- | :---: | :---: |
> | **Linear Model** | **$0.502$** | $0.740$ |
> | **Neural Network (MLP)** | $0.452$ | **$0.799$** |
>
> The results show that while the neural network achieves a higher *in-sample* $R^2$ (0.799 vs. 0.740), its *cross-validated* $R^2$ is notably lower than the linear model's (0.452 vs. 0.502). This indicates that the non-linear neural network model is overfitting the training data and generalizes more poorly than the simple linear model. These results strongly suggest that a linear model is not significantly mis-specified and is **a suitable choice for this problem**. In addition, the reasonableness of linear models is further justified by **the outstanding performance of our methods in more than 50 tasks**.
>
> As you suggested, we have added these new results to Appendix G.2.1 of the revised paper to further solidify our results. Thank you for the suggestion.
>
>
>
> ---
>
> ### **Question**:
>
> > **[W2].** There are no theoretical guarantees for the preference feedback setting in Section 4.2. While the authors motivate FedPOB-Pref with inspirations from duelling bandits and federated learning, only experimental evidence is provided for its performance.
>
> ### **Response**:
>
> We thank the reviewer for this constructive comment. You are correct that our FedPOB-Pref algorithm has only been proven effective empirically and currently does not have theoretical guarantees. We agree this is an important direction and plan to try to derive theoretical guarantees for FedPOB-Pref in future work.
>
> More specifically, the federated loss function we have adopted in FedPOB-Pref is inspired by Acar et al. (2021), in which **the theoretical convergence of this loss function has been proved**. On the other hand, the prompt selection policy of FedPOB-Pref is inspired by previous works on dueling bandits (Bengs et al., 2022; Verma et al., 2024), which has **proved the convergence of regret under a normal MLE loss function**. It is an interesting future direction to derive a complete theoretical convergence of our FedPOB-Pref algorithm by combining the analysis from these two lines of works, and we have added the corresponding discussions to Section 7 of the revised paper.

---

> > ### Author Response · Authors · 2025-11-21
> > **Response to Reviewer uKNR [Part 2 / 4]**
> >
> > ### **Question**:
> >
> > > **[W3].** The preference feedback setting is constrained to the BTL model which has known limitations. Formulation where the pairwise comparison $P(p_{t,1} \succ p_{t,2})$ is modelled directly would make the approach more applicable and general.
> >
> > ### **Response**:
> >
> > Thank you for this suggestion. We agree that exploring different preference models is a valuable direction. Our choice of the Bradley-Terry-Luce (BTL) model was deliberate, as it is **a common and standard assumption in the dueling bandits and preference-learning literature**. It provides a principled and well-studied method to connect stochastic, binary preference outcomes to an underlying latent reward function.
> >
> > Furthermore, **the BTL model has been widely adopted in the LLM field as well**. For example, this assumption is behind the majority of successful alignment methods, such as RLHF (Bai et al., 2022, Ouyang et al., 2022) and DPO (Rafailov et al., 2023, Azar et al. 2024). In these applications where real human preference feedback is used, the BTL assumption may be easily violated. Therefore, **the immense practical success of these alignment methods** provides strong justification for **the reasonableness and robustness of the BTL model**.
> >
> > In our work, **the strong empirical results for FedPOB-Pref (e.g., Table 3)**, where our method consistently and significantly outperforms the baselines, **suggest that this modeling choice leads to effective methods in practice**.
> >
> > That said, we agree with the suggestion that exploring methods which directly model pairwise comparisons is a highly promising direction for future work. We have added these discussions to the revised paper (Section 7).
> >
> > **References:**
> > - Bai et al., Training a Helpful and Harmless Assistant with Reinforcement Learning from Human Feedback. arXiv 2022
> > - Ouyang et al., Training Language Models to Follow Instructions with Human Feedback. NeurIPS 2022
> > - Rafailov et al., Direct Preference Optimization: Your Language Model is Secretly a Reward Model. NeurIPS 2023
> > - Azar et al., A General Theoretical Paradigm to Understand Learning from Human Preferences. AISTATS 2024
> >
> >
> > ---
> >
> > ### **Question**:
> >
> > > **[Q1].** Could the authors provide concrete examples to the motivation of federated learning for prompt optimisation?
> >
> > ### **Response**:
> >
> > Thank you for this question regarding the practical motivation for federated prompt optimization. We agree that concrete examples help clarify the motivation. Here are two scenarios where our framework would be directly applicable:
> >
> > - **Healthcare Diagnosis:** Different hospitals may use LLMs to assist physicians by generating diagnosis suggestions or summarizing patient histories. Each hospital needs to optimize prompts for these tasks. However, these prompts or the resulting data may contain sensitive patient information, which cannot be shared directly with other institutions due to privacy regulations. In this case, the hospitals can adopt our federated prompt optimization framework to collaboratively learn better prompts more efficiently, sharing only model parameters rather than the private data itself.
> > - **On-Device Assistants:** Mobile phone users might adopt LLMs locally on their device (or via a cloud service) for various personalized tasks, requiring them to optimize their own prompts. These prompts could relate to personal emails, messages, or schedules. Users may want to collaborate to accelerate this optimization (i.e., find the best prompts faster), yet they are naturally unwilling to share their personal prompts or interaction history due to privacy concerns. Our federated framework allows these users to benefit from collective experience without exposing their private information.
> >
> > Thank you for this suggestion. We have added these scenarios to the Introduction section of the revised paper (page 1, highlighted in blue) to provide further motivation for our work.

---

> > > ### Author Response · Authors · 2025-11-21
> > > **Response to Reviewer uKNR [Part 3 / 4]**
> > >
> > > ### **Question**:
> > >
> > > > **[Q2].** Does FedPOB work asynchronously? Is there any point when an agent has to wait for a weight update from the Central Server? ... Can multiple updates happen from Agent 1 without any from Agent 2? How would such an extreme setting affect performance?
> > >
> > > ### **Response**:
> > >
> > > We thank the reviewer for this insightful question. The asynchronous setting you have pointed out is indeed a practical and important problem setting.
> > >
> > > **The focus of our work was to simultaneously address three major challenges in federated prompt optimization**: (1) the black-box nature of proprietary LLMs, (2) the need for high sample efficiency, and (3) enabling privacy-preserving collaboration with a theoretically guaranteed benefit from more agents. Therefore, in this paper, we have not considered the additional challenge of asynchronous communication.
> > >
> > > We agree that extending our framework to the asynchronous setting is a valuable future direction. We will explore this in future work, potentially drawing inspiration from protocols for asynchronous communication in federated bandits, such as the one proposed by Li et al. (2022). We have added this as a promising direction for future work in our revised paper (Section 7).
> > >
> > > **References:**
> > > - Li et al. Asynchronous upper confidence bound algorithms for federated linear bandits. AISTATS 2022.
> > >
> > > ---
> > >
> > > ### **Question**:
> > >
> > > > **[Q3].** In line 243 the authors claim that in many practical applications only preference feedback is available or it is more applicable. What are exactly these applications?
> > >
> > > ### **Response**:
> > >
> > > Thank you for asking for clarification. Here are a couple of practical applications where preference feedback is more natural and reliable than numeric scores:
> > >
> > > - **Creative Writing & Style Alignment:** A writer may want to use an LLM to help draft an article that aligns with their own unique creative preference or writing style. The user needs to provide a detailed prompt describing this preferred response style. In this case, **it is very difficult to use a numeric score to quantify how "aligned" the generated article is with the user's subjective preference**. In contrast, asking the user to simply compare a pair of articles generated from two different prompts is often more acceptable, reliable, and less cognitively demanding. In this scenario, different writers could use our federated prompt optimization framework to more efficiently find the best prompts for their tasks, without sharing their private and personalized style prompts.
> > > - **Text-to-Image Generation:** A similar application arises when users want to optimize prompts for a text-to-image generative model. It is notoriously **difficult to obtain a single, meaningful numeric score to assess how much a generated image aligns with a user's complex aesthetic preference or mental model**. Therefore, asking for pairwise comparisons is a much more practical feedback mechanism. Users can then employ our federated prompt optimization framework to collaborate and more efficiently find the best prompts without sharing their personal prompts or generated images.
> > >
> > > Thank you for your suggestion. We have added these applications to the paper (Section 3.2) as further motivation for our FedPOB-Pref algorithm.
> > >
> > > ---
> > >
> > > ### **Question**:
> > >
> > > > **[Q4].** How does the acquisition logic (Line 3-4 in Algo 3) relate to previous works on Duelling Bandits? For example, it is similar to BAI version of the algorithm presented in [1]. Would the algorithm work with other approaches from the literature?
> > >
> > > ### **Response**:
> > >
> > > Thank you for this sharp observation. The acquisition logic in our FedPOB-Pref algorithm (Lines 3-4 in Algo 3) **is indeed inspired by a classic and theoretically grounded arm selection strategy from the dueling bandits literature**. The works of Bengs et al. (2022) and Verma et al. (2024) have shown that for both linear dueling bandits and neural dueling bandits, **this arm selection strategy leads to small regrets in theory**.
> > >
> > > You are correct that the selection of the second prompt in FedPOB-Pref (Line 4 of Algo 3) is similar to the BAI version of the algorithm presented in [1]. Specifically, the BAI algorithm from [1] aims to achieve **exploration** by encouraging the selection of a response with large **uncertainty**. Similarly, the second prompt in our FedPOB-Pref is selected by considering both exploitation and exploration, in which **exploration** is also achieved by encouraging the selection of a prompt with large **uncertainty** (as seen in the UCB-style bonus term in Line 4 of Algo 3). This is an excellent connection, and we have revised the paper (Appendix B) to add this discussion related to [1].
> > >
> > > We also agree that it is an interesting future direction to explore whether other acquisition approaches from the dueling bandits literature could be successfully adapted to our federated framework.
> > >
> > > [1]: Sample-Efficient Alignment for LLMs, Liu et al. (2024)

---

> > > > ### Author Response · Authors · 2025-11-21
> > > > **Response to Reviewer uKNR [Part 4 / 4]**
> > > >
> > > > ### **Question**:
> > > >
> > > > > **[Q5].** In Section 4, does the number of samples observed and used for learning increases with the number of agents? Is the batch size fixed per agent or per iteration? For a consistent evaluation, it would be valuable to show performance as a function of number of samples seen by all the agents.
> > > >
> > > > ### **Response**:
> > > >
> > > > Thank you for this question about the experimental evaluation. We will clarify this below.
> > > >
> > > > In our experimental setting, each agent observes **one single sample in each iteration**. This means the total number of new samples collected per iteration (i.e., the "batch size" for that iteration) is **equal to the number of agents**. For example, in the 10-agent setting, 10 new samples are observed across the federation in each iteration. You are correct that the total number of samples seen by all agents increases with the number of agents.
> > > >
> > > > **1. Performance As A Function of the Total Number of Samples**
> > > >
> > > > As you suggested, to evaluate performance as a function of the number of samples seen by all agents, we have additionally plotted the best reward (after 50 iterations, averaged across all Instruction-Induction tasks) as a function of the **total number of samples collected by all agents**. To provide a more comprehensive view, we additionally scale the number of agents to **25** and **100**, enabling us to examine the performance in larger-scale federated settings. The figure and corresponding table are displayed below:
> > > > [See figure here](https://postimg.cc/PN3gp5gw)
> > > > | # of Agents  | 1     | 3     | 10    | 25    | 100   |
> > > > |--------------|-------|-------|-------|-------|-------|
> > > > | # of Samples |50     |150    |500    |1250   |5000   |
> > > > | Best Rewards | 0.7637|0.7977 | 0.8068|0.8101 |0.8150 |
> > > >
> > > > As the number of agents increases, the marginal gain **gradually diminishes**. This suggests a **logarithmic-like scaling curve** where performance gains eventually saturate
> > > >
> > > > **2. Justification for Reporting Average Performance Per Agent**
> > > >
> > > > However, in our experiments, we reported **the average performance per agent** as a function of iterations (e.g., in Figure 2 and 3). We chose this metric because it directly characterizes **how much performance gain an individual agent obtains on average by joining the federation.**
> > > >
> > > > For example, plotting performance vs. iterations allows for a direct comparison of:
> > > > 1. The performance of a **single agent** after 50 iterations (and thus 50 personal queries).
> > > > 2.  The performance of an **agent participating in a 10-agent federation** after 50 iterations (and thus also 50 personal queries).
> > > >
> > > > Our results (e.g., Tables 1, 2 and 3, Figures 1 and 2) demonstrate the federated agent (2) achieves much higher scores, demonstrating the benefit of collaboration for **the same per-agent query cost**. We believe this provides a clear characterization of the benefit of federated prompt optimization.
> > > >
> > > > ---
> > > >
> > > > Thank you once again for your insightful comments. We hope our explanations and new results have clarified your questions.

---

> > > > > ### Comment · Reviewer_uKNR · 2025-11-25
> > > > >
> > > > > Thank you for the detailed explanations and revisions of the paper accordingly. I believe these were important adaptations to make to clarify the paper's contributions. I maintain my evaluation and score.

---

> > > > > > ### Author Response · Authors · 2025-11-26
> > > > > >
> > > > > > We sincerely appreciate the reviewer’s encouraging assessment and positive feedback. We are pleased that the thorough revisions have effectively addressed all the raised points and, as the reviewer noted, have clarified the manuscript's contributions. Thank you for maintaining your initial high evaluation.

---

### Official Review · Reviewer_4GM4 · 2025-10-30

**Soundness:** 2
**Presentation:** 2
**Contribution:** 2
**Rating:** 4
**Confidence:** 4

**Summary:**

The paper introduces FedPOB, a sample-efficient, privacy-preserving framework for black-box federated prompt optimization built on multi-armed bandits, addressing three deployment constraints at once: no gradient/parameter access, high query cost, and multi-user collaboration. It proposes two algorithms: FedPOB (a federated LinUCB that shares/aggregates linear bandit statistics instead of raw data) and FedPOB-Pref (a federated dueling bandit variant that aggregates parameters with dynamic regularization to handle noisy preference feedback efficiently). Experiments on Instruction Induction and BBH with API LLMs show consistent gains over baselines and monotonic improvements as more agents participate, while exposing a controllable performance-vs-communication trade-off via a thresholded sync rule.

**Strengths:**

1. The problem this paper study is interesting and important. It provides a score based methods and a preference based method.
2. Extensive experiments show that the proposed method outperforms baselines.

**Weaknesses:**

1. Why the rewards are decreasing as the communication rounds increase is Figure 4.
2. The reviewer is curious which experiment shows and compares the sample efficiency as this is a main assertion from the paper.
3. What is the unique contribution of the paper or the unique challenge that this solves in the FL scenarios. It looks like the paper is making a sophisticated algorithm to its FL version.

**Questions:**

See weaknesses.

---

> ### Author Response · Authors · 2025-11-21
> **Response to Reviewer 4GM4 [Part 1 / 2]**
>
> We are grateful for the reviewer's thorough and detailed review. Your insightful comments have significantly contributed to improving our paper.
>
> ---
>
> ### **Question**:
>
> > **[W1].** Why the rewards are decreasing as the communication rounds increase is Figure 4.
>
> ### **Response**:
>
> Thank you for this question, and we apologize for the confusion. We would like to clarify that Figure 4 **actually shows that the rewards are decreasing as the communication rounds decrease**.
>
> The horizontal axis in Figure 4 represents the communication **threshold $D$** (from line 8 of Algorithm 1), rather than the number of communication rounds. A larger $D$ (moving to the right on the x-axis) leads to **fewer** communication rounds, because it imposes a stricter condition on starting a communication round (see line 8 of Algorithm 1). To illustrate this, we report below the number of communication rounds corresponding to different values of $D$ across 50 iterations of FedPOB (averaged over all Instruction-Induction tasks).
>
> | $D$                     | 0    | 10    | 100   | 300 | 1000 |
> |-------------------------|------|-------|-------|-----|------|
> | **# Comm. Rounds (3 Agents)**  | 50   | 12    | 4     | 2   | 1    |
> | **# Comm. Rounds (10 Agents)** | 50   | 11.13 | 3.99  | 2   | 1    |
>
> Therefore, Figure 4 shows that **as the number of communication rounds decreases** (i.e., as the communication threshold $D$ increases), **the rewards are reduced accordingly**.
>
> We have added these results in the revised paper (Appendix G.3)
>
> We have revised the paper to add this table and make this interpretation clearer (Appendix G.3 and Section 5). Thank you for bringing this to our attention.
>
> ---
>
> ### **Question**:
>
> > **[W2].**  The reviewer is curious which experiment shows and compares the sample efficiency as this is a main assertion from the paper.
>
> ### **Response**:
>
> We thank the reviewer for this question, as sample efficiency is a key assertion of our paper.
>
> Firstly, Tables 1, 2, and 3 all demonstrate our sample efficiency. These tables show that **given the same number of samples (i.e., query budget) per agent** (i.e., 50 queries), our algorithms consistently achieve larger final scores than the baseline methods.
>
> In addition, Figures 2, 3, and 5 further validate the high sample efficiency of our FedPOB and FedPOB-Pref algorithms. These figures, which plot average reward against the number of iterations (i.e., samples per agent), show that **to achieve the same average reward,** our algorithms with more agents (e.g., 10 agents) **require significantly fewer samples/iterations** than with fewer agents (e.g., 1 agent). This directly demonstrates that the sample efficiency of our algorithms is consistently improved as the number of agents increases.
>
> We have revised the paper (Sections 4.1 and 4.2) to clarify this. Thank you for pointing this out.

---

> > ### Author Response · Authors · 2025-11-21
> > **Response to Reviewer 4GM4 [Part 2 / 2]**
> >
> > ### **Question**:
> >
> > > **[W3].** What is the unique contribution of the paper or the unique challenge that this solves in the FL scenarios. It looks like the paper is making a sophisticated algorithm to its FL version.
> >
> > ### **Response**:
> >
> > Thank you for this question, which allows us to clarify our unique contribution. The primary challenge we address is not simply federating an existing algorithm, but creating a unified framework to simultaneously tackle a unique **three-way-dilemma** in prompt optimization: the need to handle (1) **black-box** model access, (2) the demand for high **sample efficiency** due to query costs, and (3) the requirement for **privacy-preserving collaboration**.
> >
> > Our paper makes unique contributions by solving this combined challenge for the two primary types of user feedback:
> >
> > 1.  **For Score Feedback (FedPOB):** The challenge is to create a practical federated system that is both sample-efficient and communication-efficient, while also providing guaranteed benefits from collaboration. Our contribution is **FedPOB**, which adapts the Federated Linear UCB algorithm. We solve the practical challenge by designing a communication-efficient protocol (Algorithm 1) that, as shown in our ablation study (Figure 4), **maintains strong performance even with infrequent communication**. Crucially, our framework inherits the **theoretical guarantee** that **performance improves with more participating agents**, a non-trivial and core contribution for this application.
> > 2.  **For Preference Feedback (FedPOB-Pref):** This challenge is technically distinct and more complex. A simple "federated version" of a dueling bandit algorithm is **not possible**. The unique technical hurdle is that the parameter estimation for linear dueling bandits **does not have a closed-form solution**. This limitation, which is a key technical challenge, prevents the simple and effective parameter aggregation strategy that we used in FedPOB. This forces a move to gradient-based optimization. However, existing Federated approaches here (like FLDB-GD/OGD) create a new problem: a severe trade-off between high communication cost and poor performance. Our unique contribution is **FedPOB-Pref**, a novel algorithm that solves this specific dilemma. Instead of simply aggregating gradients, **we propose to aggregate model parameters and introduce a dynamic regularization technique**, which we have shown to significantly improve the performance of FedPOB-Pref (Figure 6).
> >
> > Therefore, our paper's contribution is in fact not "making a sophisticated algorithm to its FL version." It is a novel, unified framework that **provides the first effective solution to the 3-way-dilemma** (black-box, sample-efficiency, privacy) **for both score and preference feedback**. For scores (FedPOB), we provide a practical, communication-efficient algorithm with theoretical backing. For preferences (FedPOB-Pref), we overcome a significant technical barrier (the lack of a closed-form solution) to create a new algorithm that is empirically superior to existing federated methods in both performance and communication efficiency.
> >
> > ---
> >
> > Thank you once again for your thoughtful comments. We hope our new results and clarifications could help improve your evaluation of our paper.

---

### Official Review · Reviewer_CFze · 2025-11-01

**Soundness:** 3
**Presentation:** 3
**Contribution:** 3
**Rating:** 6
**Confidence:** 4

**Summary:**

This paper proposes an interesting algorithm, FedPOB, to allow distributed agents to collaborate on prompt optimization. The setting is that all the agents share the same task they want to solve but may have different prompt sets, and they want to collaborate together to help each of them find their optimal prompts. The algorithm works by having each agent use a linear model of an embedded prompt to predict the score of the solution, and training the linear model in a federated averaging-like manner. The prompts in each iteration for each agent are chosen to maximize the linear model's UCB to balance exploration and exploitation. The paper demonstrates natural scaling law such as improved performance with more agents and improved empirical performance compared to prior prompt optimization methods.

**Strengths:**

- The paper looks at prompt optimization in the setting where there are distributed agents. This is a very timely paper which looks at a problem that I think is going to become more important.
- The methods are simple and easy to understand; I can imagine them working in a wide variety of settings
- Thorough experimental evaluation. Experiments across GPT and Qwen show that the conclusions apply to different models. Experiments across tasks and datasets show wide applicability across tasks.
- Follows the typical FL protocol, which makes it seem realistic in setting and sounds productionizable
- The extension to the preference learning setting is good for improving the applicability of the method.

**Weaknesses:**

- The setting is strange. Heterogeneity is only in the prompt set but not in the tasks? I would actually rather the setting be flipped: heterogeneity in tasks (x,y pairs) but prompt set is the same. Distributed agents will probably encounter different sets of prompts, while their prompt sets would probably be the same.
- It may be better to add some ablations using nonlinear critics. There's no real reason to limit ourselves to linear models practically is there?
- Should probably discuss the Private Evolution line of work. It has a similar idea: use scores to iterate on black box LLM prompts. Including a discussion of this work and how it is related is likely best.

Lin, Zinan, et al. "Differentially private synthetic data via foundation model apis 1: Images." arXiv preprint arXiv:2305.15560 (2023).

Xie, Chulin, et al. "Differentially private synthetic data via foundation model apis 2: Text." arXiv preprint arXiv:2403.01749 (2024).

Lin, Zinan, et al. "Differentially private synthetic data via apis 3: Using simulators instead of foundation model." arXiv preprint arXiv:2502.05505 (2025).

Some follow ups of Private Evolution also look at the federated setting.

Hou, Charlie, et al. "Private federated learning using preference-optimized synthetic data." arXiv preprint arXiv:2504.16438 (2025).

**Questions:**

- I think my main concern is with the setting of this paper. Why would we have heterogeneous prompts instead of heterogeneous task datasets?

---

> ### Author Response · Authors · 2025-11-21
> **Response to Reviewer CFze [Part 1 / 3]**
>
> We sincerely thank the reviewer for their insightful feedback and constructive suggestions that have helped us improve the quality of the paper.
>
> ---
>
> ### **Question**:
>
> > The setting is strange. Heterogeneity is only in the prompt set but not in the tasks? I would actually rather the setting be flipped: heterogeneity in tasks (x,y pairs) but prompt set is the same.
>
> > Why would we have heterogeneous prompts instead of heterogeneous task datasets?
>
>
> ### **Response**:
>
> We appreciate the reviewer’s thoughtful comment regarding the problem setting. You raise a valid point about the nature of heterogeneity in federated prompt optimization. Below, we clarify the motivation behind our design choice of heterogeneous prompt sets and provide additional experimental results addressing your suggestion for heterogeneous tasks.
>
> **1. Justification for Heterogeneity in the Prompt Set**
>
> We specifically chose a setting with heterogeneous prompt sets **to better align with the privacy constraints and practical workflows of federated prompt optimization**. In standard prompt optimization frameworks (e.g., Lin et al., (2024b); Zhou et al., (2023)), the search space is typically constructed by rephrasing initial seed prompts. In a federated context, enforcing a homogeneous (identical) prompt set across all agents **would require the central server to generate a global prompt domain and broadcast it to every agent**.
>
> However, in many realistic scenarios, **agents may have proprietary local contexts or specific prompt variations they prefer to keep private**; they may not be willing to adopt a public, shared list of prompts or expose their specific search space to the server. By allowing each agent to maintain its own local prompt set (generated via local rephrasing), we ensure that agents are not forced to share their specific prompt candidates, thereby preserving a higher degree of privacy.
>
> We have added explicit discussions of this motivation in the revised paper (Section 2, highlighted in blue).
>
> **2. Additional Experiments with Heterogeneity in Tasks**
>
> We agree with your observation that a setting with a shared prompt set but heterogeneous tasks is also practically relevant and represents a valuable test case. Following your suggestion, we conducted an additional experiment where **all agents share the same prompt set (containing 500 prompts), but their tasks (i.e., reward functions) are heterogeneous**. To simulate this task heterogeneity, we modified the underlying reward function for each agent by adding distinct Gaussian noise to the original function values, resulting in different reward functions for different agents.
>
> We evaluated this setting across all 29 Instruction-Induction tasks using `gpt-3.5-turbo`. The results ([see figure here](https://postimg.cc/2Vf98zfT)) demonstrate that **FedPOB continues to perform robustly under this heterogeneous-task scenario**. Even when the agents are optimizing against heterogeneous reward functions, our algorithm successfully leverages collaboration to improve overall performance. This confirms that **FedPOB is adaptable to different types of federated heterogeneity**, whether in the prompt space or the task definition.
>
> We have added these results to the revised paper (Appendix G.5, highlighted in blue).

---

> > ### Author Response · Authors · 2025-11-21
> > **Response to Reviewer CFze [Part 2 / 3]**
> >
> > ### **Question**:
> >
> > > It may be better to add some ablations using nonlinear critics. There's no real reason to limit ourselves to linear models practically is there?
> >
> > ### **Response**:
> >
> >
> > Thank you for your suggestions. We agree that including more ablation experiments using non-linear models is a great point. To compare our linear bandit–based approach with non-linear alternatives for federated black-box prompt optimization, we evaluate two families of neural bandit baselines across 29 Instruction-Induction tasks: **federated neural bandits** (score feedback) and **federated neural dueling bandits** (preference feedback).
> >
> > **1. Score Feedback Baseline**
> > We compare against **Federated Neural Bandit** (Dai et al., 2023), a Neural-UCB–based federated bandit method. Since the original work does not include prompt optimization and its network architecture is incompatible with high-dimensional features (768-d embeddings), we adapt the neural network architecture from **APOHF** (Lin et al., 2024a), which uses the same embedding model as ours and ensures a fair, architecture-aligned comparison.
> >
> > **2. Preference Feedback Baseline**
> > For preference feedback, we extend **APOHF** (Lin et al., 2024a), a neural dueling bandit method for prompt optimization, to the federated setting. To ensure consistency and fairness, we adopt the **FedAvg** aggregation protocol, following Federated Neural Bandit (Dai et al., 2023).
> >
> > **3. Results**
> > We report the best rewards across all 29 Instruction-Induction tasks under both score-feedback and preference-feedback settings, comparing **FedPOB** ([See figure here](https://postimg.cc/cv6m3W8m)) and **FedPOB-Pref** ([See figure here](https://postimg.cc/vDW5xB2Q)) with their corresponding non-linear bandit baselines.
> >
> > Across both feedback types, compared with our **FedPOB** and **FedPOB-Pref**, **the federated neural bandit and neural dueling bandit baselines fail to scale effectively**. Their performance does **not** consistently improve as the number of agents increases. This may be due to the non-convexity of neural networks combined with data heterogeneity: local neural updates diverge across agents, making it difficult for standard federated averaging to converge to a stable global model in a small number of iterations. This in fact also aligns with theoretical results, since **federated neural bandit (Dai et al., 2023) is not guaranteed to achieve better performance with more agents**. In contrast, the performance of federated linear bandits is theoretically guaranteed to improve as the number of agents increases, which is one of our major motivations for adopting linear models.
> >
> > These new results have been added Appendix G.2.3.

---

> > > ### Author Response · Authors · 2025-11-21
> > > **Response to Reviewer CFze [Part 3 / 3]**
> > >
> > > ### **Question**:
> > >
> > > > Should probably discuss the Private Evolution line of work. It has a similar idea: use scores to iterate on black box LLM prompts. Including a discussion of this work and how it is related is likely best.
> > >
> > > ### **Response**:
> > >
> > > Thank you for pointing out this relevant line of work on "Private Evolution". We agree that it shares a similar high-level goal: using scores to guide iterative queries to black-box LLMs in a privacy-conscious setting. These works are related but solve the problem from a different and complementary perspective:
> > >
> > > - **Privacy Mechanism:** The "Private Evolution" line of work (Lin et al., 2023; Xie et al., 2024; Lin et al., 2025) primarily focuses on achieving privacy through **Differential Privacy (DP)**, often with the specific goal of synthetic data generation. Their aim is to provide formal DP guarantees for the outputs. In contrast, our paper focuses on **Federated Learning (FL)** as the privacy-preserving paradigm. In our framework, privacy is achieved by *not sharing raw data* (such as sensitive prompts or scores) among agents, who instead collaborate by only sharing aggregated model parameters.
> > > - **Federated Setting:** The follow-up work (Hou et al., 2025) also addresses a federated setting. However, it proposes using preference-optimized *synthetic data* as a mechanism for private federated learning. This is a different approach from our framework, where agents collaborate directly (via parameter sharing) on the prompt optimization task itself, rather than generating synthetic data as an intermediate step.
> > > - **Optimization Framework:** Our approach is built upon the **multi-armed bandit (MAB)** framework. This is specifically chosen to tackle the challenge of sample efficiency in prompt optimization, as MABs are inherently designed to efficiently manage the exploration-exploitation trade-off in black-box settings. This provides a different technical foundation compared to the evolutionary strategies in the cited papers.
> > >
> > > We appreciate you bringing this connection to our attention. We have added a discussion to our Related Work section (Appendix B, highlighted in blue) to situate our MAB-based federated approach relative to this DP-based evolutionary approach.
> > >
> > > ---
> > >
> > > Thank you again for your constructive feedback. We hope our clarifications and additional results could enhance your opinion of our work.

---

### Official Review · Reviewer_2qph · 2025-11-03

**Soundness:** 3
**Presentation:** 3
**Contribution:** 3
**Rating:** 4
**Confidence:** 3

**Summary:**

This paper introduces a federated framework (FedPOB) for prompt optimization for agents in decentralized settings by casting prompt selection as a linear contextual bandit over fixed prompt embeddings. The core idea of this framework is that Multiple agents collaborate by exchanging only lightweight sufficient statistics instead of raw data. The framework supports both numeric-score feedback (FedPOB, a federated LinUCB) and pairwise-preference feedback (FedPOB-Pref, a federated dueling bandit with reduced communication). This method is theoretically justified and with bandwidth efficiency (message overhead of $O(d^2)$), where the synchronization is event-triggered once the threshold D is reached; in each round, agents upload their increments and receive aggregated globals. Empirical results show that this method is sample-efficient and performance-effective over recent baselines on diverse tasks and could scale to a medium to large number of agents.

**Strengths:**

1. This method is effective with closed-weight/API LLMs, where we only needs scalar scores or pairwise preferences. No gradient information is required, and there is no raw data shared across agents.

2. The linear-bandit pooling lets multiple agents discover good prompts in a fast manner; the framework also has an event-triggered synchronization system that balances the communication frequency and updates with a threshold D, allowing many local trials between rounds, and performance improves as agent number increases.

3. The design is simple and can work with models in any scale with low-bandwidth requirements. The message size is $O(d^2)$, where d is usually no more than 1e4 scale.

**Weaknesses:**

1. The experiments use GPT-3.5-turbo, GPT-4o-mini, and Qwen3-235B on Instruction Induction and BBH—both are LLMs, which are primarily generation-style evaluations. There's no encoder-only (e.g., RoBERTa/GLUE) study, so applicability to pure encoder tasks isn't demonstrated. If this is by-design short-handed, the author should acknowledge this limitation.

2. Limited evaluation on tasks with mixture of models in diverse scales. For instance, what will the performance look like if some models are significantly smaller (thus less capable) than others (i.e., DeepSeek-R1-671b with OPT-125m)?

3. FedPOB models expected reward as linear in a fixed prompt embedding. The paper does not test what happens if the model is mis-specified (i.e., non-linear reward surfaces) nor compare against non-linear bandits (e.g., kernelized UCB, neural bandits). As a result, it's unclear how performance degrades when the embedding is misspecified or interactions are non-linear.

4. While the bandwidth cost is low, the framework is communication-heavy, with each trigger requiring two communications across many agents. The main content of the paper focuses primarily on task performance rather than efficiency. The author should provide experiments with end-to-end wall-clock latency/throughput.

**Questions:**

1. The author claims in lines 80-81 that, with more agents involved, performance improves. Is there any scaling law/ patterns?

2. Can this method scale to more than 10 agents, and does this method assume the ability of the agents is similar (as questioned also in weakness 2)?

3. The author assumes the preference feedback is generated by the Bradley-Terry-Luce (BTL) model, which is strong and arbitrary. Does this assumption actually hold under a multi-agent setting? What happens if this assumption is violated?

---

> ### Author Response · Authors · 2025-11-21
> **Response to Reviewer 2qph [Part 1 / 5]**
>
> We are grateful to the reviewer for their thorough and careful reading of our paper and for providing detailed and insightful feedback.
>
> ---
>
> ### **Question**:
>
> > **[W1].** The experiments use generative LLMs and generation-style evaluations. There's no encoder-only (e.g., RoBERTa/GLUE) study, so applicability to pure encoder tasks isn't demonstrated.  If this is by-design short-handed, the author should acknowledge this limitation.
>
> ### **Response**:
>
> Thank you for your suggestion. The reviewer is correct that our experiments focus on generative LLMs. **This focus is intentional**.
>
> Our work is primarily motivated by the significant real-world challenges of optimizing prompts for **powerful, proprietary, black-box generative models**. This is the setting where the combined challenges of black-box (API-only) access, high query costs (sample efficiency), and user privacy are most acute. **We therefore focused our evaluation on this important setting** where our proposed framework offers a unique solution.
>
> We agree that applying a similar approach to encoder-only models represents an interesting, but **distinct, problem setting**. For many traditional encoder tasks (e.g., GLUE), white-box model access is often available, making methods such as fine-tuning or soft-prompt tuning more commonly adopted. Our framework is specifically designed for improving the performance of **black-box LLMs** where those methods are not feasible. Adapting our method to such encoder-only tasks is indeed an interest topic for future research.
>
> In addition, we'd like to clarify that some of the tasks adopted in our current experiments are in fact classification tasks, which are similar to GLUE. For example, *Formal Fallacies* (from BBH dataset) is a binary classification task requiring the agent to predict whether a given argument is deductively valid or invalid; *Sentiment* (from the Instruction Induction dataset) is likewise a binary classification task in which the LLM predicts whether a statement is positive or negative.
>
> As you suggested, we have revised the paper to discuss this limitation (Section 7, highlighted in blue), in order to clarify our specific scope and acknowledge that applying our black-box optimization framework to encoder-only tasks is a promising area for future work.
>
> ---
>
> ### **Question**:
>
> > **[W2].**  Limited evaluation on tasks with mixture of models in diverse scales. For instance, what will the performance look like if some models are significantly smaller (thus less capable) than others (i.e., DeepSeek-R1-671b with OPT-125m)?
>
> > **[Q2].** Does this method assume the ability of the agents is similar (as questioned also in weakness 2)?
>
> ### **Response**:
>
> Thank you for your interest in how our method behaves when the agents have heterogeneous capabilities. To examine this, we added an experiment adopting **three LLMs with different capabilities** and evaluating FedPOB on **two mixed-model groups** across all Instruction-Induction tasks.
>
> - **Group 1:** GPT-3.5-turbo + Qwen3-235B + GPT-5-nano
> - **Group 2:** Llama-3.2-1B + Qwen3-235B + GPT-5-nano
>
> For the 3-agent setting, we directly mix the three models. For the 10-agent setting, we include four weaker agents (GPT-3.5-turbo or Llama-3.2-1B) and three stronger agents.
>
> **Results**
> Since different LLMs exhibit different score distributions, instead of averaging the rewards across all agents, we report the best reward of the mid-strength agent (Qwen3-235B, [See figure here](https://postimg.cc/WhTdJkbF)) and the strongest agent (GPT-5-nano, [See figure here](https://postimg.cc/NLw6pLfS)) separately after 50 iterations.
>
> **• Group 1: Performance steadily improves as the number of agents increases.**
> Although GPT-3.5-turbo is weaker than the other two LLMs ([See GPT-3.5-turbo results here](https://postimg.cc/SXk00Wrx)), it still contributes meaningfully to the federation. This demonstrate that FedPOB can effectively handle heterogeneous agents.
>
> **• Group 2: Performance does not increase monotonically when more agents are added.**
> We find that **llama-3.2-1B performs extremely poorly** on the prompt-optimization task ([See Llama-3.2-1B results here](https://postimg.cc/9DPSBDh3)). As a result, when such significantly weaker agents exist in the federation, the performance no longer improves with more agents, because their noisy local updates can negatively affect the stronger models.
>
> Overall, these results show that **FedPOB does not require agents to have similar capabilities**, moderately weaker models can contribute useful exploration and improve performance. But **extremely weak models may hinder performance** when they participate in the optimization process.
>
> Thank you again for pointing out this interesting ablation study. We have added these results and discussions to the paper (Appendix G.1, highlighted in blue) after revision.

---

> ### Author Response · Authors · 2025-11-21
> **Response to Reviewer 2qph [Part 2 / 5]**
>
> ### **Question**:
>
> > **[W3].** FedPOB models expected reward as linear in a fixed prompt embedding. The paper does not test what happens if the model is mis-specified (i.e., non-linear reward surfaces) nor compare against non-linear bandits (e.g., kernelized UCB, neural bandits). As a result, it's unclear how performance degrades when the embedding is misspecified or interactions are non-linear.
>
> ### **Response**:
>
> We chose linear models to optimally balance expressiveness, efficiency, and theory (lines 161-192): **(a) Effectiveness via Embeddings:** Rich pre-trained embeddings capture complex semantics, allowing simple linear functions to approximate rewards effectively. This is validated by our strong results across 53 tasks. **(b) Efficiency:** Linear models ensure simple algorithms and lightweight $O(d^2)$ communication. **(c) Theoretical Guarantees:** Unlike neural bandits, our linear framework provides rigorous guarantees that collaboration consistently improves performance as more agents are added.
>
> Below we firstly (1) justify the reasonableness of linear models in our problem, then (2) demonstrate the robustness of our method against mis-specification, and finally (3) show that our methods outperform non-linear bandits.
>
> **1. Reasonableness of Linear Models**
>
> To empirically justify the reasonableness of adopting linear models, we conducted a new study to determine **if the mapping from prompt embeddings to their performance scores is reasonably approximated by a linear model**, or if a non-linear model offers a significant advantage.
>
> **Experimental Setup:**
> - **Objective:** To compare the predictive performance of a linear regressor versus a non-linear neural network regressor.
> - **Data:** For each of the 53 tasks in our study, we used the full set of 500 prompt embeddings as the input features ($X$) and their corresponding validation scores as the target variable ($y$).
> - **Linear Model:** We used L2-regularized linear regression (Ridge Regression). This model directly corresponds to the linear model used by our algorithms.
> - **Non-linear Model:** We used a Multi-Layer Perceptron (MLP) regressor with L2 regularization. To ensure fair comparison, we adopted the same MLP architecture as APOHF (Lin et al., 2024a) which used the same embedding model as our work.
>
> **Evaluation Protocol:**
> To ensure a fair and robust comparison in the high-dimensional setting, we evaluated both the L2-regularized linear model and the L2-regularized neural network using K-fold cross-validation (CV) with $K=5$. The L2 regularization hyperparameter for each model was tuned via a nested CV loop exclusively on the training data within each outer fold. We report two metrics here:
> 1.  **Cross-Validated $R^2$ ($R^2_{\mathrm{CV}}$)**: The primary metric for generalization and predictive power on unseen data, calculated from the out-of-sample predictions on the held-out validation folds.
> 2.  **In-Sample (Training) $R^2$**: The $R^2$ score averaged across the training folds, used to assess the degree of overfitting.
>
> **Results：**
> We averaged the $R^2$ scores across all 53 tasks, shown in the table below.
>
> | Model | $R^2_{\text{CV}}$ (Predictive Power) | $R^2$ (In-sample Fit) |
> | :--- | :---: | :---: |
> | **Linear Model** | **$0.502$** | $0.740$ |
> | **Neural Network (MLP)** | $0.452$ | **$0.799$** |
>
> The results show that while the neural network achieves a higher *in-sample* $R^2$ (0.799 vs. 0.740), its *cross-validated* $R^2$ is notably lower than the linear model's (0.452 vs. 0.502). This indicates that the non-linear neural network model is overfitting the training data and generalizes more poorly than the simple linear model. These results strongly suggest that **a linear model is a suitable choice for this problem** and is not significantly mis-specified. In addition, the reasonableness of linear models is further justified by **the outstanding performance of our methods across the 53 tasks** (e.g., Tables 1 and 2).
>
> We have added these new results to Appendix G.2.1 of the revised paper.
>
> **[Response to W3 Continues Below]**

---

> > ### Author Response · Authors · 2025-11-21
> > **Response to Reviewer 2qph [Part 3 / 5]**
> >
> > **[Response to W3 Continued from above]**
> >
> > **2. Our Methods Are Robust Against Embedding Mis-specification**
> >
> > To evaluate the robustness of our method under embedding mis-specification, we conducted an ablation study where we deliberately distort the prompt embeddings. Specifically, we truncate each $768$-dimensional embedding vector by **removing its last $768/2=384$ dimensions** and then renormalize the resulting vectors. This produces a **mis-specified scenario** as the reviewer has suggested.
> >
> > The results ([see figure here](https://postimg.cc/V5ZTg9qM)) show that:
> > 1. **For 1-agent and 3-agent settings**, the performance remains nearly unchanged compared to the original results in Figure 2.
> > 2. **For the 10-agent setting**, the distorted embedding leads to a small performance drop, indicating that FedPOB becomes slightly more sensitive to model mis-specification when more agents are involved.
> > 3. **Across all settings**, the overall trend is preserved: increasing the number of agents generally improves performance.
> >
> > These findings suggest that **FedPOB maintains relatively stable behavior even under substantial embedding mis-specification**, and the benefits of more participating agents continue to hold.
> >
> > We have added these new results to Appendix G.2.2.
> >
> >
> > **3. Comparison with Non-linear Bandit Baselines**
> >
> > To compare our linear bandit–based approach with non-linear alternatives for federated black-box prompt optimization, we evaluate two families of neural bandit baselines across 29 Instruction-Induction tasks: **federated neural bandits** (score feedback) and **federated neural dueling bandits** (preference feedback).
> >
> > **(1) Score Feedback Baseline**
> > We compare against **Federated Neural Bandit** (Dai et al., 2023), a Neural-UCB–based federated bandit method. Since the original work does not include prompt optimization and its network architecture is incompatible with high-dimensional features (768-d embeddings), we adapt the neural network architecture from **APOHF** (Lin et al., 2024a), which uses the same embedding model as ours and ensures a fair, architecture-aligned comparison.
> >
> > **(2) Preference Feedback Baseline**
> > For preference feedback, we extend **APOHF** (Lin et al., 2024a), a neural dueling bandit method for prompt optimization, to the federated setting. To ensure consistency and fairness, we adopt the **FedAvg** aggregation protocol, following Federated Neural Bandit (Dai et al., 2023).
> >
> > **(3) Results**
> > We report the best rewards across all 29 Instruction-Induction tasks under both score-feedback and preference-feedback settings, comparing **FedPOB** ([See figure here](https://postimg.cc/cv6m3W8m))  and **FedPOB-Pref** ([See figure here](https://postimg.cc/vDW5xB2Q)) with their corresponding non-linear bandit baselines.
> >
> > Across both feedback types, compared with our **FedPOB** and **FedPOB-Pref**, **the federated neural bandit and neural dueling bandit baselines fail to scale effectively**. Their performance does **not** consistently improve as the number of agents increases. This may be due to the non-convexity of neural networks combined with data heterogeneity: local neural updates diverge across agents, making it difficult for standard federated averaging to converge to a stable global model in a small number of iterations. This in fact also aligns with theoretical results, since **federated neural bandit (Dai et al., 2023) is not guaranteed to achieve better performance with more agents**. In contrast, the performance of federated linear bandits is theoretically guaranteed to improve as the number of agents increases, which is one of our major motivations for adopting linear models.
> >
> > These new results have been added Appendix G.2.3.
> >
> > **Conclusion:**
> > These new findings together justify that **the linear model is a reasonable and robust choice in our problem, and performs better than non-linear bandits**.

---

> > > ### Author Response · Authors · 2025-11-21
> > > **Response to Reviewer 2qph [Part 4 / 5]**
> > >
> > > ### **Question**:
> > >
> > > > **[W4].**  While the bandwidth cost is low, the framework is communication-heavy, with each trigger requiring two communications across many agents. The main content of the paper focuses primarily on task performance rather than efficiency. The author should provide experiments with end-to-end wall-clock latency/throughput.
> > >
> > > ### **Response**:
> > >
> > >
> > > Thank you for raising this important point regarding communication efficiency. To begin with, we'd like to note that the communications in our frameworks are managed by **a central server**. As a result, for each trigger (i.e., communication round), every agent only needs to send its local parameters to the central server, who then broadcasts the aggregated parameters back to the agents. This reduces communication since the agents do not need to communicate with each other.
> > >
> > > To better demonstrate our communication efficiency, we present additional results below to show that the total number of triggers (i.e., communication rounds) and the total throughput are small for both FedPOB and FedPOB-Pref. For **throughput**, we simply measure **the total number of parameters exchanged between agents and the server over 50 iterations**
> > >
> > > **1. Score Feedback Setting (FedPOB)**
> > >
> > > In FedPOB, we use the communication threshood $D$ to limit unnecessary communication rounds, and only start a communication round if an agent has collected enough information（line 213-215). The table below reports the total number of communication rounds for different values of $D$ over 50 iterations of FedPOB (averaged across all Instruction-Induction tasks). To further quantify **throughput**, we compute **the total number of parameters exchanged** between the agents and the server over 50 iterations.
> > >
> > > | $D$                     | 0    | 10    | 100   | 300 | 1000 |
> > > |-------------------------|------|-------|-------|-----|------|
> > > | **# Comm. Rounds (3 Agents)**  | 50   | 12    | 4     | 2   | 1    |
> > > | **# Comm. Rounds (10 Agents)** | 50   | 11.13 | 3.99  | 2   | 1    |
> > > | **Throughput (3 Agents)**  |169.0M|40.56M|13.52M|6.75M|3.37M|
> > > | **Throughput (10 Agents)** |563.2M|125.4M|44.94M|22.53M| 11.26M|
> > >
> > > As expected, the frequency of communication and total throughput both decline sharply as the communication threshold $D$ increases. Figure 4 demonstrates that even when $D$ is raised to **300 or 1000**—limiting communication to **merely two rounds or a single round**—our method remains **highly robust**. It exhibits only a negligible reduction in reward, despite the substantial decrease in communication overhead.
> > >
> > >
> > > **2. Preference Feedback Setting (FedPOB-Pref)**
> > >
> > > In the preference-feedback setting, we similarly evaluate communication efficiency by examining both the number of communication rounds and total throughput. In addition to our standard **FedPOB-Pref**, we also test another variant of FedPOB-Pref in which we allow one communication trigger every three iterations, referred to as **FedPOB-Pref (Less Comm.)**. The results for 3 agents and 10 agents are displayed in the two tables below:
> > >
> > > |**3 Agents**|**Best rewards**|**Comm.Times**|**Throughput**
> > > |---------------|---------------|--------------|-----------|
> > > |**FedPOB-Pref**              |**0.8145**|50    |169.1M    |
> > > |**FedPOB-Pref (Less Comm.)** |0.7814     |**16**|**54.11M**|
> > > |**FLDB-GD** (baseline)                  |0.7959    |1500  |175.3M    |
> > > |**FLDB-OGD** (baseline)                |0.7687    |50    |169.0M    |
> > >
> > > |**10 Agents**|**Best rewards**|**Comm.Times**|**Throughput**
> > > |---------------|---------------|--------------|-----------|
> > > |**FedPOB-Pref** |**0.8482**|50    |563.9M|
> > > |**FedPOB-Pref (Less Comm.)** |0.7882|**16**|**180.4M**|
> > > |**FLDB-GD** (baseline)     |0.8244|1500  |584.4M|
> > > |**FLDB-OGD** (baseline)   |0.8123|50    |563.5M|
> > >
> > > These results demonstrate the following:
> > > - **FedPOB-Pref outperforms FLDB-OGD by 5.95%**, while using the **same** number of communication rounds.
> > > - **FedPOB-Pref matches FLDB-GD’s performance while requiring only 1/30 of its communication rounds.**
> > > - **FedPOB-Pref remains robust even under much stricter communication constraints.**
> > > When we allow only one communication trigger every three iterations, the total number of communication rounds and the overall throughput are reduced by **68%**, while the performance declines by **only 4.06% and 7.07%** for 3 and 10 agents, respectively. This demonstrates that FedPOB-Pref provides a reliable option in scenarios where communication resources are highly limited or expensive.
> > >
> > >
> > > Thank you again for your suggestion. We have added these new results and discussions to the paper (Appendix G.3).

---

> > > > ### Author Response · Authors · 2025-11-21
> > > > **Response to Reviewer 2qph [Part 5 / 5]**
> > > >
> > > > ### **Question**:
> > > >
> > > > > **[Q1].**  The author claims in lines 80-81 that, with more agents involved, performance improves. Is there any scaling law/ patterns?
> > > >
> > > > > **[Q2].** Can this method scale to more than 10 agents?
> > > >
> > > >
> > > > ### **Response**:
> > > >
> > > > We appreciate the reviewer’s questions regarding the scalability of our methods to more agents and potential scaling laws. To address your concerns, we have conducted additional experiments with a larger number of agents.
> > > >
> > > > We adopted the Instruction-Induction tasks (29 tasks), and **increased the number of agents to 25 and 100** under the same experimental setting (using `gpt-3.5-turbo`). The table below reports the **final aggregated scores across all 29 tasks** after 50 iterations (extension of Table 5).
> > > >
> > > > | # of Agents        | 1     | 3     | 10    | 25    | 100   |
> > > > |--------------|-------|-------|-------|-------|-------|
> > > > | Best Rewards | 0.7637 | 0.7977 | 0.8068 | 0.8101 | 0.8150 |
> > > >
> > > > We also plotted these results as a curve, which provides a clearer **visualization of the scaling law**: **[see figure here](https://postimg.cc/YGkcycC5)**. We also plot the **average reward curves across iterations** (extension of Figure 2): **[see figure here](https://postimg.cc/LhvRpZjr)**.
> > > >
> > > > These results suggest a clear scaling pattern:
> > > > 1. **Our method is indeed able to scale to a substantially larger number of agents** ($N=100$).
> > > > 2. The performance of FedPOB **improves consistently and monotonically** as the number of agents increases.
> > > > 3. As the number of agents increases, the marginal gain **gradually diminishes**. This suggests a **logarithmic-like scaling curve** where performance gains eventually saturate, which is a desirable and expected property in large-scale collaborative learning systems.
> > > >
> > > >
> > > >
> > > > ---
> > > >
> > > > ### **Question**:
> > > >
> > > > > **[Q3].** The author assumes the preference feedback is generated by the Bradley-Terry-Luce (BTL) model, which is strong and arbitrary. Does this assumption actually hold under a multi-agent setting? What happens if this assumption is violated?
> > > >
> > > > ### **Response**:
> > > >
> > > > Thank you for this insightful question. Our choice of the Bradley-Terry-Luce (BTL) model was deliberate, as it is **a common and standard assumption in the dueling bandits and preference-learning literature**. It is in fact not an arbitrary choice; it provides a principled and well-studied method to connect stochastic, binary preference outcomes to an underlying latent reward function.
> > > >
> > > > Furthermore, **the BTL model has been widely adopted in the LLM field as well**. For example, this assumption is behind the majority of successful alignment methods, such as RLHF (Bai et al., 2022, Ouyang et al., 2022) and DPO (Rafailov et al., 2023, Azar et al. 2024). In these applications where real human preference feedback is used, the BTL assumption may be easily violated. Therefore, **the immense practical success of these alignment methods** provides strong justification for **the reasonableness and robustness of the BTL model**.
> > > >
> > > > In our work, this assumption is indeed not guaranteed to hold for the different agents as the reviewer has alluded to. However, **the strong empirical results for FedPOB-Pref (e.g., Table 3)**, where our method consistently and significantly outperforms the baselines, **suggest that this modeling choice leads to effective methods in practice**.
> > > >
> > > > **References:**
> > > > - Bai et al., Training a Helpful and Harmless Assistant with Reinforcement Learning from Human Feedback. arXiv 2022
> > > > - Ouyang et al., Training Language Models to Follow Instructions with Human Feedback. NeurIPS 2022
> > > > - Rafailov et al., Direct Preference Optimization: Your Language Model is Secretly a Reward Model. NeurIPS 2023
> > > > - Azar et al., A General Theoretical Paradigm to Understand Learning from Human Preferences. AISTATS 2024
> > > >
> > > >
> > > > ---
> > > >
> > > > Thank you once again for your valuable comments. We hope our clarifications and new results could improve your evaluation of our work.

---

### Author Response · Authors · 2025-11-30
**Summary of Rebuttal and Revisions**

Dear Area Chair and Reviewers,

We sincerely thank you for the time and effort dedicated to reviewing our paper. We have carefully considered all comments and are pleased to report that we have addressed all concerns in our rebuttal. We have revised our paper accordingly, with all changes **highlighted in blue**. Below is a summary of our responses and the corresponding revisions.

* **Linearity Assumption and Comparison with Non-Linear Baselines (Reviewers 2qph, CFze, uKNR)**
    * *Question:* The reliance on linear models was questioned. Reviewers suggested testing for mis-specification and comparing against non-linear/neural bandits.
    * *Response & Revision:* We empirically justified the linear model by showing it achieves a higher Cross-Validated $R^2$ compared to an MLP (which overfits). We demonstrated robustness to mis-specification by testing FedPOB with deliberately distorted embeddings. Furthermore, we compared our methods against Federated Neural Bandits and Neural Dueling Bandits, showing that our approach scales better and outperforms them. These results are added to **Appendix G.2 (G.2.1, G.2.2, G.2.3)**.

* **Heterogeneity in Agents and Tasks (Reviewers 2qph, CFze)**
    * *Question:* Reviewers asked about performance with heterogeneous agent capabilities (mixed strong/weak models) and why we chose heterogeneous prompt sets over heterogeneous tasks.
    * *Response & Revision:*
        1.  *Agent Heterogeneity:* We added experiments mixing models (GPT-3.5, Qwen, GPT-5-nano, Llama-3.2). Results show FedPOB is robust, though extremely weak agents can hinder performance. Added to **Appendix G.1**.
        2.  *Task Heterogeneity:* We explained that heterogeneous prompt sets preserve privacy. We also added an experiment with shared prompts but heterogeneous tasks (noisy reward functions), showing FedPOB remains robust. Added to **Appendix G.5** and justified in **Section 2**.

* **Scalability and Scaling Laws (Reviewers 2qph)**
    * *Question:* Can the method scale beyond 10 agents, and is there a scaling law?
    * *Response & Revision:* We extended experiments to 25 and 100 agents. The results demonstrate a clear logarithmic-like scaling pattern where performance improves monotonically with more agents. Added to **Appendix G.4**.

* **Communication Efficiency and Latency (Reviewers 2qph, 4GM4)**
    * *Question:* Concerns were raised about communication overhead/throughput and the interpretation of the reward curve regarding communication thresholds.
    * *Response & Revision:* We clarified that Figure 4 plots the *threshold* $D$ (where higher $D$ means *fewer* rounds). We provided tables quantifying communication rounds and throughput, showing our methods are efficient. We also tested a "Less Comm." variant of FedPOB-Pref. Added to **Appendix G.3** and clarified in **Section 5**.

* **Preference Feedback Modeling and Applications (Reviewers 2qph, uKNR)**
    * *Question:* Questions regarding the validity of the BTL assumption, lack of theoretical guarantees for FedPOB-Pref, and practical applications of preference feedback.
    * *Response & Revision:*
        1.  *BTL:* We justified BTL as standard in RLHF/DPO and acknowledged the limitation.
        2.  *Theory:* We discussed the theoretical basis (Acar et al. 2021) and marked full derivation as future work (**Section 7**).
        3.  *Applications:* We detailed use cases like Creative Writing and Text-to-Image generation where scores are hard to define. Added to **Section 3.2**.

* **Scope of Evaluation (Reviewer 2qph)**
    * *Question:* The evaluation focused on generative models, lacking encoder-only tasks.
    * *Response & Revision:* We clarified that our focus is on black-box generative optimization where gradients are inaccessible. We acknowledged this scope limitation and the potential for future work on encoder tasks in **Section 7**.

* **Related Work and Comparisons (Reviewers CFze, uKNR)**
    * *Question:* Suggestions to discuss additional related work.
    * *Response & Revision:* We added a discussion distinguishing our FL-based approach from DP-based "Private Evolution" and acknowledged the connection of our acquisition logic to the BAI algorithm by Liu et al. Added to **Appendix B (Related Work)**.

* **Sample Efficiency (Reviewer 4GM4)**
    * *Question:* Request for explicit evidence of sample efficiency.
    * *Response & Revision:* We pointed to Tables 1-3 and Figures 2, 3, and 5, illustrating that federated agents achieve higher scores than single agents given the same per-agent query budget. Clarified in **Sections 4.1 and 4.2**.

* **Practical Motivation and Async Settings (Reviewer uKNR)**
    * *Question:* Requested concrete examples of federated prompt optimization and asked about asynchronous updates.
    * *Response & Revision:* We added examples (Healthcare Diagnosis, On-Device Assistants) to **Page 1 (Introduction)**. We noted asynchronous communication as a promising future direction in **Section 7**.

---

### Note · Program_Chairs · 2026-01-17
**Submission Desk Rejected by Program Chairs**

The following references in this submission do not refer to real documents and/or have major errors in bibliographic information:

 Ziyu Ye, Hao-Yang Chen, Yong-Qiang Hu, Zhen-Yu Su, Qing-An Yao, Yu-Hong Liu, Xiao-Rong Lai, and Yi-Feng Wu. Align-Pro: A principled approach to prompt optimization for llm alignment. arXiv preprint arXiv:2308.11585, 2023.